# Full hardware implementation of neuromorphic visual system based on multimodal optoelectronic resistive memory arrays for versatile image processing

Guangdong Zhou[1,7], Jie Li[2,7], Qunliang Song [3], Lidan Wang [1], Zhijun Ren[1], Bai Sun [4], Xiaofang Hu[1], Wenhua Wang[3], Gaobo Xu[3], Xiaodie Chen[5], Lan Cheng[6], Feichi Zhou [2] ✉ & Shukai Duan [1] ✉

In-sensor and near-sensor computing are becoming the next-generation computing paradigm for high-density and low-power sensory processing. To fulfil a high-density and efficient neuromorphic visual system with fully hierarchical emulation of the retina and visual cortex, emerging multimodal neuromorphic devices for multi-stage processing and a fully hardware-implemented system with versatile image processing functions are still lacking and highly desirable. Here we demonstrate an emerging multimodal-multifunctional resistive random-access memory (RRAM) device array based on modified silk fibroin protein (MSFP), exhibiting both optoelectronic RRAM (ORRAM) mode featured by unique negative and positive photoconductance memory and electrical RRAM (ERRAM) mode featured by analogue resistive switching. A full hardware implementation of the artificial visual system with versatile image processing functions is realised for the first time, including ORRAM mode array for the in-sensor image pre-processing (contrast enhancement, background denoising, feature extraction) and ERRAM mode array for near-sensor high-level image recognition, which hugely improves the integration density, and simply the circuit design and the fabrication and integration complexity.

In the conventional machine vision system, image sensors are separately connected to the memory and processing units by adopting the von Neumann computing architecture[1]. Conventional image sensing usually occurs in the analogue domain, after which the sensory data is further converted to digital signal through analogue-digital converters (ADCs), then transferred to the memory and processing units[2–4]. Thus, sensory data processing based on the conventional architecture suffers from long-distance communication from sensory units to processing units with a limited data transfer rate[5]. The massive amount of raw and redundant sensory data conversion and transmission causes

[1]College of Artificial Intelligence, Chongqing Key Laboratory of Brain-inspired Computing and Intelligent Chips, Key Laboratory of Luminescence Analysis and Molecular Sensors (Ministry of Education), Southwest University, Chongqing 400715, China. [2]School of Microelectronics, Southern University of Science and Technology, Shenzhen 518055, China. [3]Faculty of Materials and Energy, Southwest University, Chongqing 400715, China. [4]Frontier Institute of Science and Technology, Xi'an Jiaotong University, Shanxi 710049, China. [5]Department of Electrical and Electronic Engineering, The University of Hong Kong, Hong Kong 999077, China. [6]State Key Laboratory of Silkworm Genome, College of Sericulture, Textile and Biomass Sciences, Southwest University, Chongqing 400715, China. [7]These authors contributed equally: Guangdong Zhou, Jie Li. ✉e-mail: zhoufc@sustech.edu.cn; duansk@swu.edu.cn

high energy consumption, high latency, and limited communication bandwidth, all of which are key issues for the state-of-the-art machine vision hardware system. Thus, an efficient and highly integrated sensory computing architecture for various intelligent tasks is desirable to solve the sensory processing efficiency and latency bottlenecks in the machine vision system[6–9].

In contrast, the human visual system including the retina and visual cortex, outperforms the conventional imagers and machine vision systems, particularly in unique architecture and processing schemes with high integration density and energy efficiency[9,10]. The retina holds a hierarchical biostructure composed of rod/cone photoreceptors, bipolar cells, amacrine cells, and ganglion cells, which all play significant roles in visual sensing and pre-processing, such as photoreception, feature extraction, temporal contrast enhancement, and denoising, etc.[11,12]. The pre-processed visual information by the retina will be further transmitted to the visual cortex for high-level processing tasks such as learning, memory, inference, and recognition[8,9]. This processing scheme of in-retina sensing and pre-processing and high-level visual information processing in the visual cortex can effectively reduce the redundant data transfer and conversion between sensor and processing units, alleviating the latency and efficiency bottlenecks and further simplifying the hardware design complexity with a higher integration level[7,10].

Inspired by the human visual cortex, neuromorphic visual systems with in-memory computing architecture have been proposed by integrating the image sensors with near-sensor resistive random access memory (RRAM) array-based post-processor, which can improve the integration density and energy efficiency from a certain degree through adopting high-density in-memory computing architecture and shortening the distance between sensing and processing units[13,14]. However, the large amount of redundant sensory data collected from the image sensors still brings computing burden and hardware design complexity to the post-processor based on the resistive random access memory, which puts higher demands on the accuracy and processing capability of the RRAM-based in-memory processing system design.

To further improve the artificial visual system performance, milestone breakthroughs of in-sensor computing systems have been demonstrated[5–10,13–21]. The retina-inspired in-sensor computing devices such as optoelectronic resistive random access memory (ORRAM) that enable executing in-sensor image pre-processing at the front end are proposed to emulate the advanced sensing and pre-processing functions in the human retina, allowing for effectively reducing the unnecessary sensory data and extracting the feature information, and further simplifying the hardware complexity and improving the learning efficiency and accuracy of neural network in the post-processor[6,7,10,13]. However, the reported in-sensor image pre-processing and post-processing are still based on the simulation instead of the real hardware system demonstration. Moreover, currently, the ORRAM arrays or phototransistor arrays can only complete the limited image pre-processing functions, lacking multiple image pre-processing capabilities and high-level processing capabilities. Then the two-dimensional material WSe$_2$-based transistor arrays with tuneable photo responsibility are proposed for the ultrafast machine vision system with new computing architecture, which can complete $3 \times 3$ handwritten letter recognition, image encoding, and decoding in the sensor units[9]. Although an advanced hardware demonstration based on the WSe$_2$-based transistor arrays has been realised for such image processing applications, the multiple subpixels-constituted one-pixel core in the proposed computing architecture restricts the image resolution and the complex and flexible image processing applications. Therefore, the current challenges met in the in-sensor computing systems based on emerging devices are mainly concentrated on the realisation of versatile image processing functions towards flexible application scenarios, originating from the restrictions of device functionalities, computing architecture, and algorithm designs[6,7,19–21]. Thus, the realisation of multi-functionalities in image processing is highly desirable for an advanced visual system towards practical application. More importantly, a full hardware system demonstration that fully emulates the hierarchical processing architecture in the human visual system, including both the retina and visual cortex, has not been realised to the best of our knowledge.

Herein, an emerging multimodal modified silk fibroin protein (MSFP)-based multimodal resistive memory arrays based full hardware implemented visual system with versatile image processing functions is first demonstrated in this work, fully simulating the retina cells to conduct image pre-processing and the visual cortex to complete high-level image processing. The novel multimodal MSFP-based memory exhibits two modes of operations including both analogue optical resistive switching mode (ORRAM mode) and analogue electrical resistive switching mode (ERRAM mode). Especially, the device in ORRAM mode presents unique optically controlled positive photoconductance memory (PPM) and negative photoconductance memory (NPM) effects, which allow the full optical SET and optical RESET operations and further multiple image pre-processing functions. The multimodal device allows the integration of image sensing and versatile image pre-processing and high-level processing functions. A full hardware-implemented neuromorphic visual system was built by employing two identical multimodal resistive memory arrays with different operation modes for the first time. Combined with the design of novel optical convolution algorithms, the front ORRAM mode array enables in-sensor contrast enhancement, background denoising and image erasing, and in-sensor convolutional operations for the feature extraction function. In addition, the post-processor based on the identical memory array in ERRAM mode can complete the in-memory image recognition with reduced fabrication and integration complexities. The MSFP-based resistive memory array-based full hardware implemented neuromorphic system shows promising potential in the future machine vision system with multi-functionalities, high integration level, and low power consumption.

## Results

Figure 1 shows the schematic diagram of an advanced neuromorphic vision system based on MSFP-based multimodal resistive memory arrays with both ORRAM mode for image pre-processing (e.g. denoising, contrast enhancement, feature extraction, image erasing) and ERRAM mode for high-level image recognition (e.g. recognition), simulating the functions of retina cells and visual cortex. In the biological visual system, the human retina is organised into three nuclear layers and two synaptic layers, in which bipolar cells that include the off and on cells as an inner nuclear layer (INL) connect the outer nuclear layer (ONL, photoreceptors) and the ganglion cell layer (GCL)[11,12]. The images are firstly sensed by the ONL and then are pre-processed by the INL and GCL, and the pre-processed information will be finally transmitted to the visual cortex layer to complete high-level processing.

Analogous to the above human visual system, a full hardware-implemented neuromorphic vision system including two identical multimodal resistive switching memory arrays in ORRAM mode and ERRAM mode with the peripheral circuits is constructed, as shown in Fig. 1. To be more specific, the multimodal MSFP-based memory in the ORRAM mode can directly respond to the optical signals demonstrating both continuously tuneable positive photoconductance memory (PPM) and negative photoconductance memory (NPM) phenomenon according to the light intensity, which enables the implementation of in-sensor image preprocessing (e.g. image contrast enhancement, background denoising, feature extraction and image erasing). The multimodal MSFP-based memory in the ERRAM mode exhibits electrical analogue resistive switching memory behaviours, allowing for further high-level processing such as image recognition.

# Neuromorphic vision chip based on multimodal resistive memory arrays

**Fig. 1 | Neuromorphic vision chip based on multimodal resistive memory arrays with ORRAM mode array for in-sensor image preprocessing and ERRAM mode array for high-level image recognition.** An advanced neuromorphic vision system based on MSFP-based multimodal resistive memory arrays with both ORRAM mode for image pre-processing and ERRAM mode for high-level image recognition, simulating the functions of retina cells and visual cortex, respectively. In the biological visual system, the human retina is organised into three nuclear layers and two synaptic layers, in which bipolar cells that include the off and on cells as an inner nuclear layer (INL) connect the outer nuclear layer (ONL, photoreceptors) and the ganglion cell layer (GCL). The images are firstly sensed by the ONL and then are pre-processed by the INL and GCL, and the pre-processed information will be finally transmitted to the visual cortex layer to complete high-level processing. The neuromorphic vision chip consists of an ORRAM mode array with NPM and PPM features for the in-sensor image pre-processing (e.g. contrast enhancement and background denoising) and in-sensor convolution for feature extraction, and ERRAM mode array with analogue resistive switching for in-memory high-level image recognition through convolutional neural network (CNN) operations.

The novel design and behaviours in the multimodal MSFP-based memory array allow for simplified array fabrication and circuit design in the fully hardware-implemented neuromorphic visual system.

A two-terminal multimodal MSFP-based resistive memory with a cell area of $200\,\mu m \times 200\,\mu m$ and a structure of $35\,nm$ Au/$100\,nm$ modified silk fibroin protein (MSFP) switching layer/$35\,nm$ Au was fabricated on the MSFP flexible substrate (Fig. 2a and Supplementary Fig. 1). The optical image of a $12 \times 12$ crossbar array with an overall area of $4.6\,mm \times 4.6\,mm$ is shown in Supplementary Fig. 1a, b. Silk fibroin protein (SFP) with a specific amino acid series can be chemically exfoliated from cocoon, showing great potential for the flexible and biocompatible RRAM device since the hydroxyl bonds and carbon-oxygen double bonds in amino acid series could provide active reaction sites for hydrogen bond reaction or polymerisation reaction to form a series of traps for electrical resistive switching[22,23]. However, the SFP-based functional films presented today cannot display optically induced resistive switching features[22–25]. Therefore, an MSFP thin film is first developed in this work by introducing polyglycerol-3 (Pg-3) and 5-6-dihydroxyindole (5-6-DHI) with rich hydroxyl bonds and carbon-oxygen double bonds to the SFP. The additives can easily form strong hydrogen bonds with the SFP and alter the secondary structures. This process helps to introduce trap states and light-tuneable secondary structure changes in the MSFP thin film for both electrical and optical switching (Supplementary Fig. 2). Figure 2b shows the synthesis route of the MSFP thin film formed by the hydrogen-bond interaction between SFP with a dialyzing time of 48 h and the Pg-3 and 5-6-DHI. The MSFP thin film shows a denser and smoother surface than the SFP thin film, showing higher thin film quality (Supplementary Fig. 3). To further examine the structural change from SFP to MSFP thin films, the Fourier transform infrared spectroscopy (FTIR) characterisations are conducted, which suggest new structures such as $-CH_2$-based chains

formed and no observation of the chemical bond of C–O located at $1165\,cm^{-1}$ in the MSFP films (Supplementary Fig. 4), indicating that the change of C–O-based bond groups possibly provide the chemical reaction sites to form a large number of hydrogen bonds among SFP, Pg-3 and 5,6-DHI. In addition, the MSFP material holds a higher dielectric constant ($40.62@1\,MHz$) (Supplementary Fig. 5a) and stronger absorption for visible light than the SFP material (Supplementary Fig. 5b), enabling the MSFP-based electronic device with good electrical and optical resistance tuneability.

The electrical resistive switching behaviours are first studied in the multimodal Au/MSFP/Au device. Figure 2c presents a tuneable analogue resistive memory behaviour under both continuous positive voltage ($0 \rightarrow 1 \rightarrow 0$ V) and negative voltage ($0 \rightarrow -1 \rightarrow 0$ V) sweepings. The device shows good cyclic endurance and non-volatile memory with a retention time of over $10^4$ s (Supplementary Fig. 6a, b). The cumulative probability of the low resistance state (LRS) and high resistance state (LRS) for 100 different devices suggests good device-to-device stability under the electrical stimuli (Supplementary Fig. 6c). The response speed of the device is illustrated in Supplementary Fig. 7, showing a fast response speed of $10\,\mu s$. In comparison with the SFP-based memory showing digital switching and abrupt set and reset process (Supplementary Fig. 8a–c), the Au/MSFP/Au memory device shows an analogue switching behaviour with non-volatile 16 multilevel resistance states as shown in Supplementary Fig. 9. Figure 2d exhibits the conductance updates with 50 pulses ($0.7$ V, $50\,\mu s$) for the long-term potentiation (LTP) process and another 50 pulses ($-0.7$ V, $50\,\mu s$) for the long-term depression (LTD) process. These programmed conductance states after LTP and LTD processes can be well maintained (Supplementary Fig. 10). These electrically tuneable LTP and LTD characteristics in the Au/MSFP/Au memory enable further in-memory computing for image recognition applications.

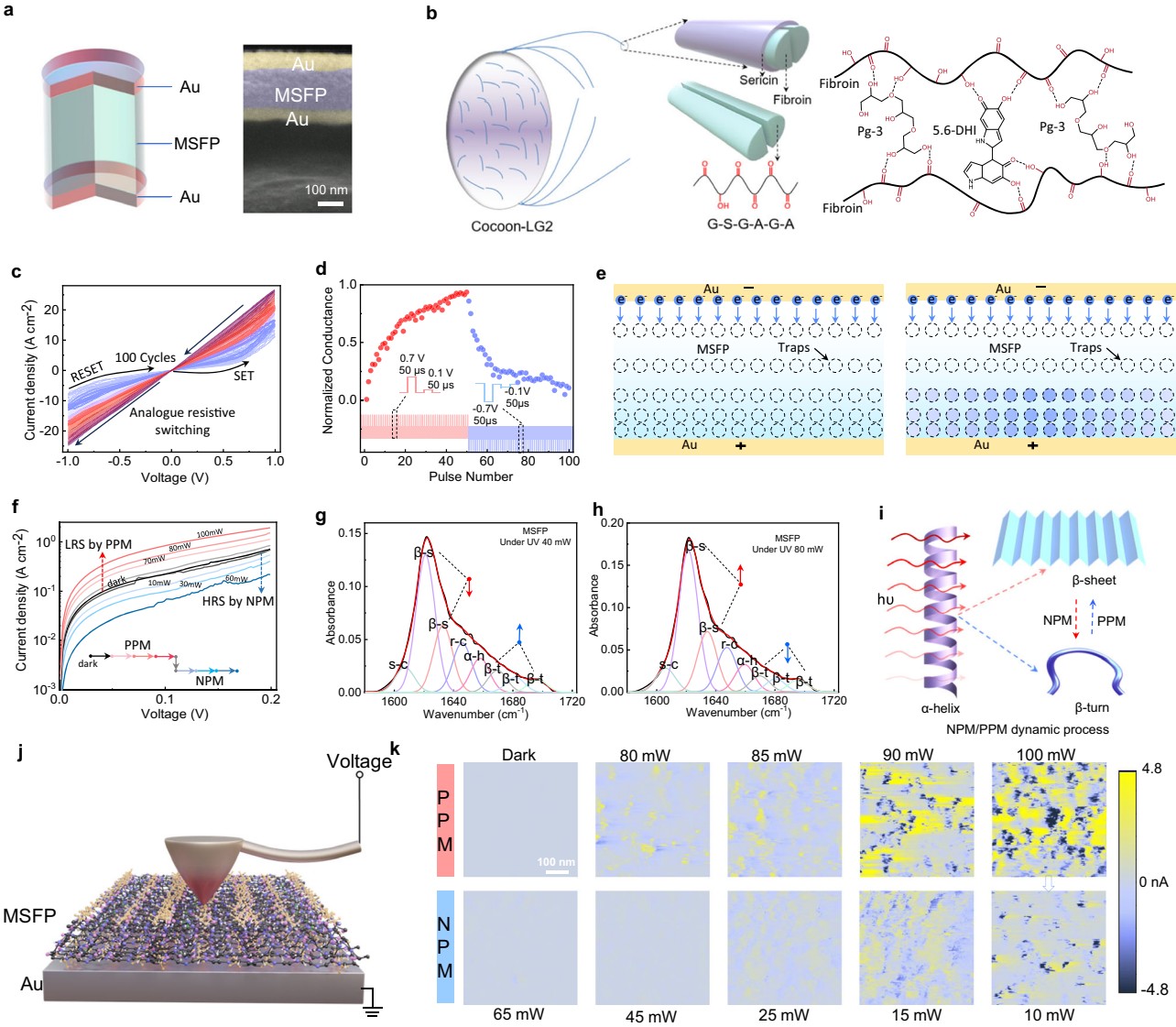

**Fig. 2 | Multimodal analogue resistive memory: electrical resistive switching (ERRAM mode) and pure optically controlled NPM/PPM switching (ORRAM mode). a** Schematic diagram of the multimodal-multifunction MSFP-based resistive memory. The right inset is a cross-sectional FE-SEM image. **b** Fibroin extracted from cocoon-LG2 and the chemical structure of modified silk fibroin protein (MSFP) formed by hydrogen-bond interactions. **c** Analogue electrical resistive switching behaviour of the Au/MSFP/Au memory in which the MSFP function switching function film is synthesised by the SFP solution, Pg-3, and 5,6-DHI. The voltage scanning rate is 0.2 V/s. **d** The long-term potentiation and depression under the continuous positive pulse stimuli (0.7 V, 50 μs) and negative (−0.7 V, 50 μs) pulse stimuli. The conductance states are readout by a read voltage pulse (0.1 V, 50 μs). **e** Switching mechanisms of analogue electrical resistive switching behaviours induced by the iteration processes of electron filling in the traps. **f** Tuneable and light intensity-dependent NPM and PPM, indicating the fully

optical set and reset processes. FTIR spectra peak fittings of the MSFP thin film secondary structures in the Amide first region under (**g**) 40 mW and (**h**) 80 mW UV light, respectively, suggesting the light illumination can alter the MSFP secondary structure. **i** Schematics of optically induced secondary structure changes leading to PPM and NPM effects. The PPM effect is mainly dominated by the net increasing amount of β-sheet secondary structures, while the NPM effect is dominated by the net increasing amount of β-turn secondary structures. **j** Measurement setup of current-sensor atomic force microscope (CS-AFM) for the Au/MSFP sample. **k** CS-AFM images of the Au/MSFP sample under a variety of light intensities. The sample exhibited obvious NPM and PPM effects when illuminated by the 10−65 mW and 80−100 mW 405 nm lasers, respectively. The light-related measurements of (**f**−**h**) and (**k**) are conducted in air ambient with relative humidity of 45% with the light exposure duration of 2.0 s.

The analogue electrical switching mechanisms in the device can be attributed to the space charge limit current (SCLC) mechanism, as shown in Fig. 2e. Under a positive voltage, the electrons are injected into the trap sites in the MSFP film causing the resistance transition from HRS to LRS, while the captured electrons were de-trapped from the defect sites under negative voltage resulting in the resistance transition from LRS back to HRS. The analogue resistive memory behaviours with multiple states are possibly dependent on the ion doping concentration such as $Li^+$ in the MSFP thin film. A suitable residual $Li^+$ ion concentration that mainly originated from the dialysis

process of the SFP preparation can introduce defect energy levels in the MSFP function film, providing suitable trap sites to store the injected electrons. The SCLC fitting of these continuous $I-V$ curves illustrates that the conduction mechanism at the voltage sweeping regions of 0 -0.05 V and 0.75 ~ 1.0 V are dominated by Mott-Gurrey law while the device conduction mechanism at 0.05 ~ 0.75 V is dominated by the transition between Ohmic conduction and Mott-Gurrey law (Supplementary Fig. 11a). According to the threshold voltage ($V_t$) corresponding to the transitions between Mott-Gurrey law-dominated region and Ohmic conduction-dominated region, dielectric constant

($\varepsilon_r$), and thickness of the MSFP thin film ($L$), the trap concentration ($N_t$) can be calculated by the equation of $N_t = \frac{2\varepsilon_0 \varepsilon_r}{eL^2} V_t$ [24,26–28]. The corresponding $N_t$ calculations illustrate that the MSFP-based memory has a high trap concentration (~$3.0 \times 10^{16}$) at HRS while it has a relatively low trap concentration at LRS for the 1st $I$–$V$ sweeping cycle (Supplementary Fig. 11b). The $N_t$ gradually decreases from ~$3.0 \times 10^{16}$ to ~$2.4 \times 10^{16}$ with the $I$–$V$ sweeping cycle increasing from 1st to 15th, indicating that the traps are gradually filled by the injected electrons. Therefore, at the very beginning state, the traps uniformly distribute in the MSFP film and then being iteratively filled by the injected electrons causing an analogue resistive switching memory behaviour. To further examine the trap concentration-based resistive switching memory behaviours, the MSFP memory devices with different Li+ concentrations and thus different trapping concentrations are fabricated to further prove the switching mechanism as shown in Supplementary Fig. 12a–c and Supplementary Table 1.

In addition to the analogue electrical resistive switching characteristics, the multimodal resistive memory exhibits fully optical-controlled resistive switching behaviours, corresponding to the ORRAM operation mode. Taking the HRS under dark as the initial state, the MSFP-based memory can be optically operated to higher resistance states by the 405 nm laser with the light intensities from 70 to 100 mW that corresponds to the positive photoconductive memory (PPM) effect, and also be optically operated to low resistance states by the 405 nm laser with the light intensities from 10 to 60 mW that corresponds to the negative photoconductive memory (NPM) effects (Fig. 2f). Before the light illumination, a voltage sweeping from 0 to 0.2 V is applied to the device under dark, corresponding to the dark resistance state. When the 405 nm light is illuminated with the intensity increased from 70 to 100 mW, the device resistance undergoes a continuous transition from the dark resistance state to lower resistance states with the increased light intensity, demonstrating the PPM effect. It is worth noting that the tuneable resistance states are non-volatile, which can be retained even after the removal of the light stimuli. Then the 405 nm light stimuli with a light intensity of 60 mW can switch off the device back from the lower resistance state to the original dark resistance state, corresponding to the optical RESET process. More importantly, the 405 nm light stimuli with increased intensities from 10 ~ 60 mW can further trigger the continuous resistance transitions from the dark resistance state to even higher resistance states, demonstrating the NPM effects, as shown in Fig. 2f. Different from the previously reported ORRAM devices with unidirectionally optical switching[10,13,19], the MSFP-based memory can be bidirectionally optical switched ON and OFF with both tuneable NPM and PPM effects, which are significant for building in-sensor pre-processing units. This optically PPM and NPM switching presents good cycle-to-cycle and device-to-device stabilities (Supplementary Fig. 13a, b).

The optical switching behaviours are possibly related to the optically induced secondary structure change in the MSFP thin films. To further reveal the dynamics of the optical switching characteristics, the MSFP's secondary structures before and after the light illuminations with different intensities are analysed through Fourier transform infrared spectroscopy (FTIR) as shown in Supplementary Fig. 4, which reveals the chemical bond vibrations (i.e., –OH, –C=O, C=C, C–OH) and the detailed information for the secondary structures in the first amide region. The FTIR results of the MSFP film before the light illumination are illustrated in Supplementary Fig. 4. The protein secondary structures that consist of the side-chain ($s$-$c$), random-coil ($r$-$c$), alpha-helix ($a$-$h$), beta-sheet ($\beta$-$s$), and beta-turn ($\beta$-$t$) were quantised in the first amide region. The concentrations of $s$-$c$, $r$-$c$, $a$-$h$, $\beta$-$s$, and $\beta$-$t$ secondary structures in the MSFP film are 8.39%, 53.09%, 14.06%, 10.75%, and 13.17%, respectively (Supplementary Table 2). For comparison, the quantification results of SFP thin film in the amide first region illustrate that the mole ratios of $s$-$c$, $r$-$c$, $a$-$h$, $\beta$-$s$, and $\beta$-$t$ secondary structures are

4.88%, 35.81%, 24.3%, 18.03%, and 16.98% in the SFP thin film (Supplementary Fig. 14a). It proves that the hydrogen-bond interaction between the SFP and the Pg-3/5-6-DHI to form the MSFP thin film can alter the protein secondary structures[24,25].

Interestingly, these secondary structures in the MSFP film can be modulated by light intensities. After the MSFP film was exposed to 80 mW UV light, the mole ratio of $\beta$-$s$ obviously increased by 11.70% while the $r$-$c$, $a$-$h$, and $\beta$-$t$ decreased by 9.86%, 24.0%, and 5.77%, respectively (Fig. 2g). When the light intensity changes from 80 to 40 mW, the mole ratio of the $\beta$-$s$ decreases from 59.30% to 55.76% while the mole ratio of $\beta$-$t$ increases from 12.41% to 14.51% (Fig. 2h), causing the NPM effect. The optically induced PPM effect can be mainly due to the increased concentration of beta-sheet ($\beta$-$s$) with layer-by-layer structures that contribute to higher conductivity, while the optically induced NPM effect can be attributed to the increased beta-turn ($\beta$-$t$) or other non-layered structures that contribute to a lower conductivity[29–31], as shown in Fig. 2i. Therefore, the increase in the $\beta$-$s$ structure (contributing to higher conductivity), along with the decrease in the $\beta$-$t$ (contributing to higher conductivity), is possibly responsible for the PPM effect observed in the Au/MSFP/Au memory. It can also be noted that when the UV light intensity changes to 60 mW, the mole ratios of the $\beta$-$s$ and $\beta$-$t$ secondary structures increase by 2.7% and 29.15%, corresponding to an enhanced NPM effect (Supplementary Fig. 14b). The comparisons of the changes in these secondary structures under the light illumination with different light intensities (40, 60, and 80 mW) are shown in Supplementary Table 3. The effects of different MSFP film thicknesses and device areas on the optical PPM and NPM switching are also investigated as shown in Supplementary Fig. 15.

To further verify the relationships between the device conductance and light intensity during NPM and PPM processes, current-sensor atomic force microscope (CS-AFM) measurements were conducted on the MSFP/Au sample under the 405 nm light illumination with different intensities (Fig. 2j). As shown in Fig. 2k, the sensor current mapping represents the conductance levels of the MSFP film within a scanning area of $500 \times 500\ nm^2$. The produced sensor current can be real-time monitored by a constant read voltage of 0.1 V applied to the Pt electrode (Au electrode is grounded) under the light illumination. Under dark, the sensing current was detected around ~0.01 nA in the selected area of the MSFP film, indicating the HRS. After the illumination of 80 mW light, an obvious local sensing current increasing that uniformly distributed in the MSPF thin film was observed in the same selected MSFP film region. With the light intensity increased from 80 to 100 mW, the local sensing current in the same region displayed an orderly enhancing tendency over the whole scanning area of the thin film, corresponding to the PPM effect in the device. After that, the MSFP thin film was exposed to a 10 mW light illumination, and the sensing current was largely suppressed and decreased from ~3.5 to ~0.8 nA, indicating the NPM effect. The NPM effect was further enhanced with the increased light intensity from 15 to 65 mW, presenting a continuously decreased current.

To systematically investigate the optically induced PPM and NPM effects, the detailed light-dosage-dependent optical switching characteristics are demonstrated in Fig. 3a. Figure 3a shows the photo-induced potentiation (PPM effect) and photoinduced depression (NPM effect) as a function of light pulse number. The photoinduced potentiation process was obtained by applying 50 light pulses (405 nm, 80 mW, 200 ms) followed by another 50 light pulses (405 nm, 40 mW, 200 ms) for the photoinduced depression process, illustrating a full optically induced analogue switching behaviour in the device. The optically tuneable PPM and NPM plasticity transitions from short-term plasticity (STP) to long-term plasticity (LTP) can be achieved in the multimodal MSFP-based device with ORRAM operation mode. A paired pulse (pulse width of 100 ms, pulse interval of 100 ms) with the light intensity varying from 70 to 100 mW was applied to the

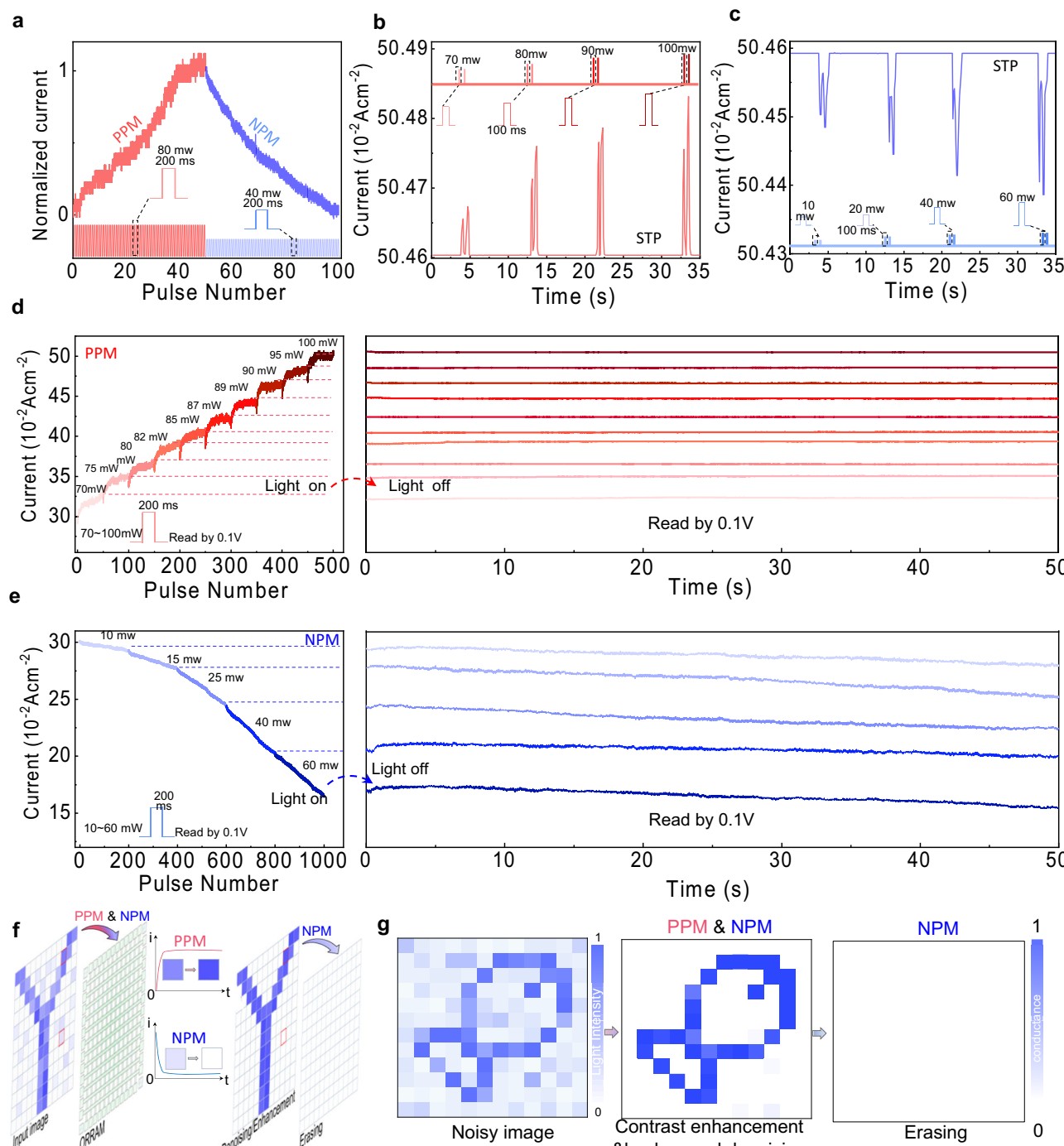

**Fig. 3 | Light tuneable NPM and PPM effects and synaptic characteristics. a** Optically induced potentiation and depression processes under the continuous 80 mW, 200 ms light pulses and 40 mW, 200 ms light pulses, respectively. **b, c** PPM and NPM STP triggered by pair pulses with light intensities ranging from 10 to 60 mW and 70 to 100 mW, respectively. **d** The PPM effect with LTP features under the increased pulse number and the varied light intensities ranging from 70 to 100 mW. **e** The NPM effect with LTP features under the increased pulse number and the varied light intensities ranging from 10 to 60 mW. All photoconductance states were read out by 0.1 V. **f** Demonstration of the image contrast enhancement and denoising through the simultaneous NMP and PPM effects, and image erasing through NPM effects. **g** A "fish" image was sensed, denoised and enhanced in situ a 12 × 12 MSFP-based multimodal memory crossbar array. The memorised and pre-processing image in the array can be further erased by light illumination.

device, indicating the PPM short-term plasticity (STP) feature (Fig. 3b). Similarly, the device displays the NPM STP feature under a series of paired-pulse stimulations (pulse width of 100 ms, pulse interval of 100 ms) with light intensity varying from 10 to 60 mW (Fig. 3c). All photoconductance states were read out by 0.1 V.

The STP to LTP transitions for both PPM and NPM occurred when the light pulse number was further increased. The left of Fig. 3d shows

the PPM effect and conductance potentiation with the increased light pulse number and an increased light intensity ranging from 70 to 100 mW (pulse number for each light intensity is 50). The PPM effect presents a growth trend with the pulse number under a higher light intensity. Overall, the conductance is increased with the light intensity level. The right of Fig. 3d demonstrates the non-volatile retention properties of the memorised states after the removal of the pulse

sequence with 50 pulses at each light intensity (70, 75, 80, 82, 85, 87, 89, 90, 95, 100 mW). The discrete photoconductance states can be well maintained after turning off the light stimuli, demonstrating the non-volatile multiple photoconductance states. The left of Fig. 3e presents the NPM effect and conductance depression with the increased pulse number at the different light intensities ranging from 10 to 60 mW (pulse number for each light intensity is 200). The device current presents a continuously decreasing trend with the increased pulse number and the light intensity. The right of Fig. 3e shows the retention of the conductance state after the removal of the pulse sequence with 200 pulses at each light intensity (10, 15, 25, 40 and 60 mW). After the removal of the light stimuli, the multiple conductance states can be well maintained. The effect of the electrical switching on the optical switching is also investigated, as shown in Supplementary Fig. 16. The comparison of our device in both optical switching and electrical switching with the devices based on other materials is shown in Supplementary Table 4.

The unique light intensity-dependent PPM and NPM effect observed in our device in ORRAM mode are highly desirable for further in-sensor image preprocessing applications, enabling high integration levels and avoiding the complex processing circuits and analogue-digital converters (ADCs) between separate sensors and processing units. The in-sensor image pre-processing functions based on the PPM and NPM effects, including the image contrast enhancement, noise reduction, and image erasing, are schematically shown in Fig. 3f. A "fish" pattern with background noise is mapped on the 12 × 12 MSFP-based memory crossbar array, outputting the responsive currents. The image mapping process is illustrated in Supplementary Fig. 17.

When the images are input into the array, the body pattern pixels with higher light intensities were enhanced through the PPM effect while the background pixels with lower light intensities were reversely suppressed by the NPM effect, resulting in the enlarged pixel signal ratio between the body pattern pixels and the background pixels. With the increased epochs (increased pulse number), the contrast is continuously enhanced (Supplementary Fig. 18), leading to an enhanced contrast of ~33 after 25 epochs. By comparison, the input image shows a light intensity ratio of ~5 between the background pixel and body pattern pixel. Therefore, a ~6 times contrast enhancement is obtained after the image preprocessing based on the ORRAM mode array (Fig. 3g). Therefore, the body pattern is enhanced by correspondingly smoothing the background noise. After completing the image pre-processing, the MSFP-based memory crossbar array could be erased by the light through the NPM effect by applying 25 light pulses of 40 mW and 200 ms, allowing for the next round of image sensing and processing (Fig. 3g).

Owing to the novel multimodal resistive switching characteristics including the non-volatile and analogue PPM and NPM optical resistive switching and electrical resistive switching in the multimodal resistive memory, a full hardware neuromorphic system containing both image pre-processing (contrast enhancement/background denoising, image erasing, and feature extraction) and high-level image recognition was built and realised for versatile in-sensor image processing applications through flexible algorithm deployment for the first time. This sensing-memory-computing integrated architecture based on novel optical/electrical MSFP-based memory arrays can not only reduce the frequent data transfer between the conventional sensor unit and the post-processor but also reduce the circuit complexity and memory fabrication complexity with largely improved high hardware integration density[7,10,19].

As shown in Fig. 4a, the neuromorphic vision hardware system is composed of an MSFP-based memory array operating in ORRAM mode, an identical MSFP-memory array operating in ERRAM mode, and the peripheral control circuits, in which the ORRAM operation mode is for in-sensor convolutional operation for image feature extractions and other image pre-processing including image contrast

enhancement/background denoising and image erasing as discussed in Fig. 3f, g, while the ERRAM mode is for the further in-memory post-processing such as image recognition (Fig. 4a). The photo of the MSFP-based multimodal-multifunction neuromorphic vision sensing hardware system is shown in Fig. 4b. The image preprocessing algorithms and a convolutional neural network (CNN) are deployed on the hardware system. The 12 × 12 ORRAM array executes the image pre-processing of contrast enhancement and background denoising and performs the first-layer optical convolutional operation in CNN for image feature extraction. The ERRAM mode array is employed to complete one convolution layer and fully connected layer computations in the CNN. Other network operations, such as max pooling layers and ReLU activation function layers, are executed within the ARM core.

Firstly, for the ORRAM mode array, image contrast enhancement/background denoising is performed according to the NPM and PPM operations in the ORRAM mode array. Utilising PPM and NPM effects, the body pattern pixels in the input image represented by relatively higher brightness are enhanced by the PPM while the background pixels with relatively lower brightness are suppressed by the NPM effect, Therefore, a contrast-enhanced image with smoothed background noise can be obtained (Fig. 4d, e). Additionally, the ORRAM array can also implement the in-sensor convolutional operation for image feature extraction (Fig. 4d). A novel optical convolution algorithm deployed on the ORRAM array, as shown in Fig. 4c, is proposed for in-sensor convolution. Each cell in the ORRAM mode array corresponds to one pixel in the input image. The ORRAM mode array can firstly sense and store the image information in the photoconductance represented using $G_{ml}$ ($G_{ml}$, $m = 1, 2, 3...n$; $l = 1, 2, 3...n$). For the optical convolution process, a reconfigurable 3 × 3 convolution kernel is first mapped into three reading voltages ($V_{11}$, $V_{12}$, $V_{13}$) at the temporal domain ($T_1$, $T_2$, $T_3$), which is then input into the ORRAM mode memory array for convolutional operations with the stored photoconductance. The output currents for every column at $T_1$, $T_2$ and $T_3$ are defined to be $I_{11}$, $I_{12}$, $I_{13}$; $I_{21}$, $I_{22}$, $I_{23}$; and $I_{31}$, $I_{32}$, $I_{33}$, respectively. The photoconductance ($G_{ml}$), reading voltage ($V_{km}$), and current ($I_{kl}$) at each ORRAM array column obey the Ohm's law. For each column of the memory array, the summation current obeys Kirchhoff's law. Therefore, the relationship of the summation current for each column, read voltage, and photoconductance can be described as follows:

$$I_{kl} = \sum_{m=1}^{3} V_{km} G_{ml} \qquad (1)$$

In this approach, a convolution kernel is divided into $T_1$, $T_2$, and $T_3$ segments. The current summation at $T_1$ is first collected and stored in the ARM core, followed by the current summation at $T_2$ and $T_3$. Then new output currents are extracted from a specific current group ($I_{11}$, $I_{22}$, $I_{33}$; $I_{12}$, $I_{23}$, $I_{34}$; $I_{13}$, $I_{24}$, $I_{33}$...) as follows:

$$I_{out-k,l} = I_{k,l} + I_{k+1,l+1} + I_{k+2,l+2} \qquad (2)$$

The employment of the novel optical convolutional operation in the ORRAM array offers a 75% reduction in time complexity compared to the traditional convolution operation method based on RRAM[32].

The convolution kernel adopted here for feature extraction is [[1/9,1/9,1/9], [1/9,1/9,1/9], [1/9,1/9,1/9]]. The feature size of the convolution kernel is 1 × 3 × 3 × 3 (depth × weight × weight × batch). For the hardware implementation, the convolution core is converted into voltage pulses and input into the ORRAM mode array through DACs. The output currents are then converted into analogue voltages through TIA, which are further converted into digital signals by ADCs and then read by ARM core master control. After the optical convolutional operation, a feature size of 12 × 12 × 3 is obtained, which further becomes 6 × 6 × 3 after the ReLu activation function and 2 × 2

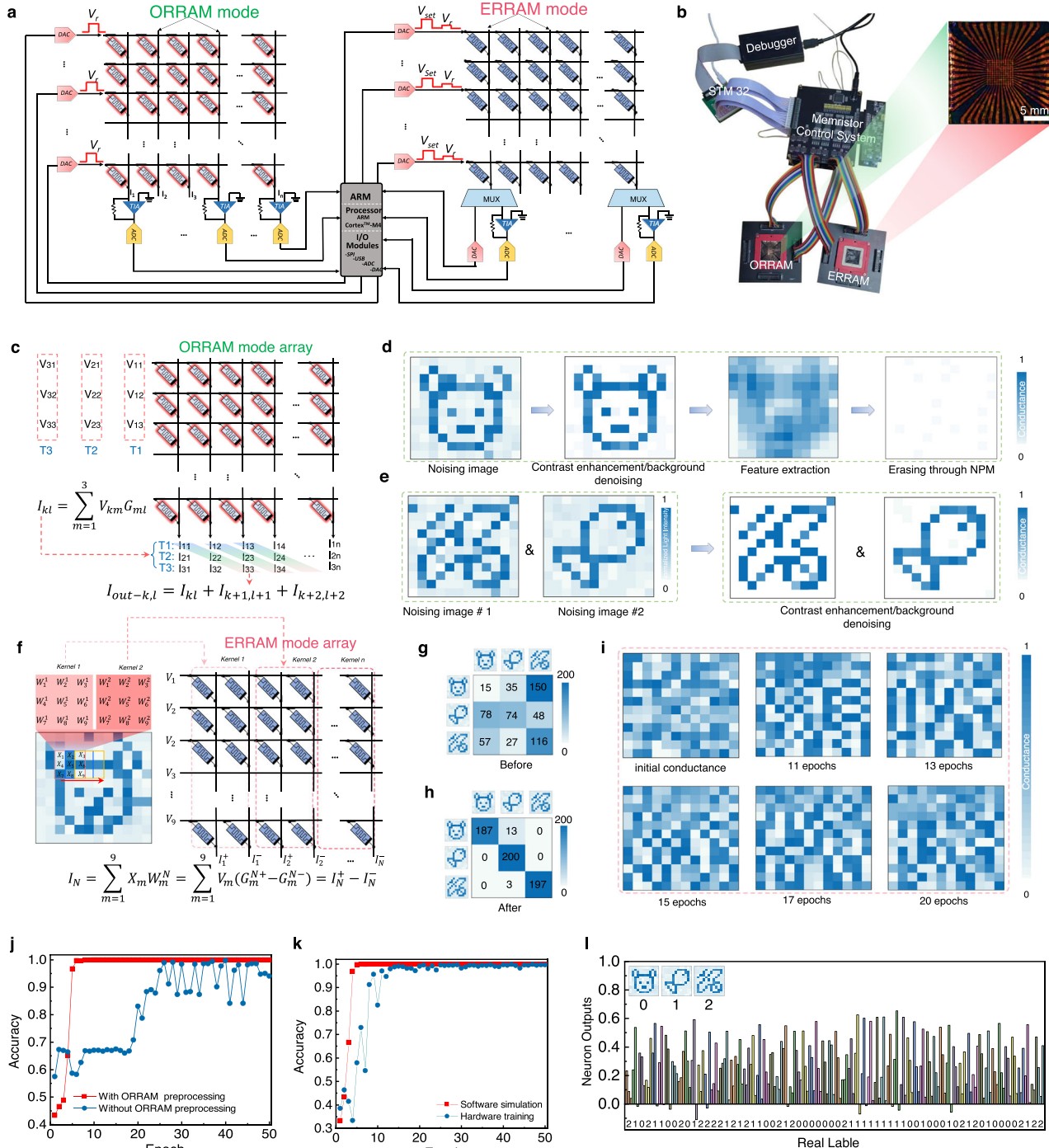

**Fig. 4 | Fully hardware implementation of the neuromorphic visual system based on the multimodal resistive memory arrays with ORRAM mode array for in-sensor pre-processing and the ERRAM mode array for near-sensor high-level processing. a** Architecture of the fully hardware-implemented neuromorphic visual system composed of an ORRAM mode array, an ERRAM mode array, a memory array peripheral control system and an STM32 system. **b** Photo of the fully hardware-implemented neuromorphic visual system. **c** In-sensor convolution operation executed in the ORRAM mode array. **d** Demonstration of in-sensor image feature extraction through in-sensor convolution operation and image erasing through NPM effects in the ORRAM mode array. **e** Demonstrations of in-sensor image denoising and contrast enhancement through PPM and NPM effects in the

ORRAM mode array. **f** Convolution implementation in the ERRAM mode array. Confusion matrices of the classifications of "bear", "fish", and "butterfly" images (**g**) before and (**h**) after 50 training epochs, yielding an accuracy of 97.3%. **i** The conductance distributions (weight maps) in the ERRAM array under the initial state and different training epochs. **j** Comparison of the image recognition results with and without the ORRAM mode array for pre-processing. **k** Comparison of the image recognition results implemented through hardware and simulation. **l** Image recognition capability of the full hardware-implemented visual system. The images of the bear, fish, and butterfly were labelled by the digit of 0, 1, and 2, respectively. These images can be recognised by our neuromorphic visual system with high accuracies.

maximum pooling operation executed in the ARM core. This newly generated feature will be further input into the ERRAM mode array-based post-processor for CNN-based image recognition.

The post-processor is composed of the identical ERRAM mode crossbar array that is aimed at high-level image processing by implementing the CNN algorithm (Fig. 4f). The extracted features by the ORRAM mode array are cached in the ARM core and then sent to the ERRAM mode array for convolution operation. Before being sent into the ERRAM mode array, $3 \times 3$ patterns in the image are sequentially flattened into column vectors. Considering the influence of interconnect resistance, the first three rows of the memory array in the $12 \times 12$ ERRAM mode array are not operated. The first six columns of the last nine rows in the ERRAM are adopted for three different convolution cores, and the last six columns are employed as a single-layer perceptron classification network with nine input and three output neurons. The output current of the different neurons can be given by the following equation:

$$I_N = \sum_{m=1}^{9} X_m W_m^N = \sum_{m=1}^{9} V_m \left( G_m^{N+} - G_m^{N-} \right) = I_N^+ - I_N^- \quad (3)$$

Where $X_m$ represents the value of input data which can be mapped to $V_m$, and $W_m^N$ denotes the weight of the neural network. The weight is represented using a pair of differential conductances $G_m^{N+}$ and $G_m^{N-}$. The $V_m$ and $G_m$ denote the read voltage applied to the MSFP memory cell and the corresponding conduction value, respectively. $I_N$ is the differential result current between two columns, $I_N^+$ and $I_N^-$. The result of the convolution calculation is denoted as $I_N$.

The convolution operation is executed with a kernel of $3 \times 3 \times 3 \times 1$, resulting in an output feature size of $6 \times 6 \times 1$. Similar to the previous activation and pooling operations, the $6 \times 6 \times 1$ feature is reduced to a feature size of $3 \times 3 \times 1$. The 9 feature values are input into a $9 \times 3$ fully connected layer to conduct classification training in the ERRAM mode array-based post-processor (Supplementary Fig. 19). The corresponding outputs are activated by the SoftMax function to generate an identified probability output. It can be noted that the above activation, pooling, and update calculations are conducted in the ARM core with all weights quantised by 3 bits. The stochastic gradient descent (SGD) optimisation algorithm and the cross-entropy loss function are used in the CNN operation[33–36]. The inexact gradient update algorithm was employed to conduct the online training[33]. The overall hardware system working flow is described in the "Method" part and Supplementary Note 7.

A demonstration of the image pre-processing and post-processing based on the ORRAM mode and ERRAM mode array hardware is illustrated in Fig. 4a. For the first step, an optical illumination of 60 mW was employed to clear the background signals in the array through the NPM effect. Then a "bear" image with background noise is input into the ORRAM array, outputting the "bear" image with enhanced contrast and smoothed background noise through ORRAM PPM and NPM effects (Fig. 4d). Moreover, the feature of the "bear" image can be extracted through the in-sensor convolutional operation. After that, the features of the "bear" can be re-cleared with light illumination through the NPM effect. To further demonstrate the capability of image pre-processing, "butterfly" and "fish" images are orderly input the ORRAM array according to the above steps (input image → output image results → image erasing), showing the reliable in-sensor pre-processing (Fig. 4e). To further prove the feasibility of the ORRAM hardware in pre-processing large image size, the MNIST handwritten digit image was tested through simulation based on the ORRAM array performances, showing a good capability in image denoising, contrast enhancement, and feature extraction (Supplementary Fig. 20). Figure 4g, h demonstrates the image recognition results based on the ERRAM mode array-based post-processor. To perform image recognition tasks, three separate training datasets consisting of 1000 images each are established for the patterns of "bear", "fish", and "butterfly". The confusion matrices of the classifications of "bear", "fish", and "butterfly" images before and after 50 training epochs are shown in Fig. 4g, h yielding an accuracy of 97.3%. Figure 4i depicts the conductance distributions (weight maps) of the $12 \times 12$ MSFP memory array during the training. After initialisation, the $12 \times 12$ MSFP memory array is in a completely random state. As the training epochs increase, the accuracy of network recognition also improves. The conductance distributions for five different training epochs (network accuracies) are presented as shown in Fig. 4i. This result also suggests the good programming capability of the MSFP memory array. To demonstrate this point, a series of voltage pulses were randomly input into the ERRAM mode memory crossbar array to evaluate the weight tuning capability (Supplementary Fig. 21a). The ERRAMs can be cyclically and repeatedly programmed into different conductive states, as shown in Supplementary Fig. 21b, c. To further demonstrate the convolutional capabilities of the ERRAM array, higher-resolution images are tested. Supplementary Fig. 21d presents the image blurring result of the "MNIST handwritten digit" image after the convolution operation.

Figure 4j, k illustrates the image recognition results of our hardware system with and without ORRAM mode memory-based pre-processing, respectively. For the hardware system recognition results, it can be noted that an accuracy of 97.3% can be obtained after 6 training epochs with ORRAM mode array-based preprocessing while a fluctuant accuracy of 95% can be achieved after 25 training epochs without ORRAM mode array-based preprocessing (Fig. 4j). The training efficiency of the CNN with ORRAM mode preprocessing is obviously higher than that without ORRAM mode pre-processing. Software simulation was conducted to prove the accuracy and effectiveness of our hardware system, showing similar recognition results (Fig. 4k). In this way, the ERRAM array-based hardware can well-recognise the image of "bear", "fish", and "butterfly" that was respectively labelled as digit 0, 1, and 2 via the operating CNN algorithm (Fig. 4l). The results of the whole hardware system indicate that the multimodal-multifunction MSFP-based memory hardware system has the capabilities to mimic the retina cells' function to conduct image pre-processing as well as to simulate the visual cortex to operate high-level image processing. By comparison, this multimodal array-based visual system shows promising potential for future in-sensing neuromorphic computing systems, exhibiting system advantages in terms of integration density and power efficiency compared with the state-of-the-art systems (Supplementary Tables 5 and 6).

We demonstrated a fully MSFP memory-implemented neuromorphic machine vision system inspired by a human visual system for high-efficiency image processing. An MSFP-based memory crossbar array with the ORRAM mode displays the NPM effect under light intensity varying from 10 to 60 mW and the PPM effect under light intensity varying from 70 to 100 mW, enabling the ORRAM mode arrays to faithfully mimic the functions of retina cells to conduct the high-efficiency image preprocessing such as contrast enhancement, feature extraction, and denoising. The developed MSFP-based memory crossbar array under the ERRAM mode presents tuneable, stable and analogue resistive switching memory behaviours, endowing the hardware system with high capability to complete high-level image processing such as image recognition. The hardware system implemented with the ORRAM mode array and ERRAM mode array exhibits high image recognition accuracy of 97.3%.

## Methods

### MSFP synthesis

A cocoon of the practical silkworm strain LiangGuang-No.2 (LG-2) was boiled in 0.02 M $NaCO_3$ solution for 30 min to move the sericin and then washed with deionized water to remove the residual $NaCO_3$. Therefore, the silk fibroin protein (SFP) cellulose was obtained after

degumming techniques. The silk fibroin protein cellulose was dissolved into 9.3 M lithium bromide (LiBr) solution at 60 °C for 3 h and then dialysed for 36, 48, or 72 h to obtain SFP precursor solution. The precursor solution was freeze-drying to obtain SFP solid to reserve its properties.

A 0.028 g freeze-drying SFP, 0. 0025 g Pg-3, and 0.0025 g 5, 6-DHI were orderly dissolved into 2 ml deionized water, ultrasonically processed for 30 s, and statically processed for 45 min to prepare a modified silk fibroin protein (MSFP) precursor that was presented black colour. A flexible MSFP substrate could be synthesised after this black precursor thermal processing at 97 °C for 3 h.

### MSFP-based memory array fabrication

The MSFP-based memory device was prepared using radio frequency (RF) magnetron sputtering and spin coating techniques. Au bottom electrode (BE) was deposited on substrates by the sputtering under the condition of 0.8 Pa Ar, 15 W, and 35 s. A ~5 nm Pt film as an adhesion layer between Au BE and substrates was considered. The MSFP-based precursor solution was spin-coated on the Au BE that was processed by plasma treatment for 60 s at 70 W. Au top electrode (TE) was fabricated by the sputtering under Ar atmosphere with a working pressure of 0.8 Pa and power of 15 W for 20 s. Thus, a multimodal memory device with an Au/MSFP/Au sandwich structure was developed.

A shadow mask with 12 strip gaps was employed to prepare the TEs and BEs. For the first step, the steel mask was covered on the MSFP substrate or quartz substrate, after which the Pt and Au film were deposited to fabricate the BEs by sputtering under an atmosphere with a working pressure of 0.8 Pa and power of 15 W for 35 s. The MSFP switching layer was prepared by spin-coating at 4500 rpm for 30 s at room temperature. Similar to the BEs processing, the TEs were fabricated by the sputtering at the same condition. After removing the steel mask, an MSFP-based memory crossbar array could be developed.

### Optoelectrical measurement and characterisation

A photoelectric integrated test platform including semiconductor analyser (B1500A, Keysight), laser (colbot, 405 nm, 0 - 100 mW) with pulse modulation units, oscilloscope (UNI-T, UPO7072Z), and probe station (Lakeshore, TTPX) was employed for the photoelectric measurements. The image patterns that irradiate to the MSFP-based crossbar array were determined by the masks with certain patterns.

The morphology, dielectric property, UV ultraviolet spectra, and infrared spectra were performed using field emission scanning electron microscopy (FE-SEM, 7100), impedance analyser, UV-vis spectrometer (SHIMADZU, UV-2700i) and Fourier transform infrared spectrometer (FTIR, Nicolet iS50), respectively. Secondary structures of the SFP and MSFP in the amide I region were obtained from the FTIR spectra peak-fitting by the software of peakfitv-4.2.

### Control system

The whole control system is composed of upper computer, STM32 system board and customised PCB board. The test script is written in Python. The upper computer sends test instructions to STM32 through serial communication to control the DAC on a customised PCB board to output voltage of different amplitude, ADC to collect voltage value, and switch the channel of the multiplexer. The 36-channel 16-bit DACs are integrated on the test board, among which each DAC can control its voltage value independently. The 24-channel 16bits ADC is integrated. The 24-channel ADC can be collected at the same time to ensure the synchronisation of the collected voltage. The customised PCB board has 24 TIAs of different ranges to adapt to the conversion of different levels of current into voltage values. All multiplexers are analogue multiplexers. Digital signals control different port switching. On-board multi-channel regulated power supply module to ensure that it can provide enough low-ripple positive and negative power supply for the whole system and improve the stability of the system.

### Hardware system workflow

A complete CNN neural network consists of one layer of ORRAM convolution, max pooling, ReLU activation function, one layer of ERRAM convolution, max pooling, ReLU activation function, and one fully connected layer. For the full hardware implementation, the ORRAM array first perceives image information, and then for the in-sensor convolution operation for image feature extraction. The values of the convolution kernels, ranging from −1 to 1, are mapped to a voltage range of −0.2 V to 0.2 V generated by the DAC and then input into the ORRAM array. The TIA on the column acts as a virtual ground, converting the incoming current into voltage, which is further collected and converted into a digital signal by the ADC. A 2 × 2 max-pooling operation and ReLU activation function are performed within the ARM core based on these extracted features. Subsequently, these features are further input into ERRAM for convolution and fully connected layer computations. Due to the ReLU activation function, all feature values become positive. When the features are input into the ERRAM array, all feature values are normalised to the range of 0 to 1, and then mapped to the voltage range of 0 to 0.2 V. These normalised feature values are converted into voltage pulses using the DAC and are input into the ERRAM. The convolution operation is performed first, with the TIAs on the columns acting as virtual grounds, collecting the currents on the columns and converting them back into voltages. Afterwards, a 2 × 2 max-pooling operation and ReLU activation function are executed within the ARM core. Finally, the processed feature values are input into the ERRAM "fully connected layer" to perform classification and output the results.

## Data availability

The data that support the plots within this paper are available from the corresponding author upon reasonable request.

## Code availability

The simulation codes used for this study are available from the corresponding author upon reasonable request.

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

## Acknowledgements

This work was supported by the Fundamental Research Funds for the Central Universities (Grant No. SWU020019), National Key Research and Development Project of China (2023YFB2806300), National Natural Science Foundation of China (Grant Nos. 62104091, 52273246, 62174074, 62274081, U20A20227, 62076207, 62076208), Guangdong Natural Science Foundation (Grant No. 2022A1515011064), Shenzhen Fundamental Research Program (Grant No. JCYJ20220530115204009), Chongqing Natual Science Foundation (2023NSCQ-LZX0142), SUSTech SME-Pixelcore Neuromorphic In-sensor Computing Joint Lab, Shenzhen Pixelcore Technology Co., Ltd, and Guangdong Provincial Engineering Research Center of 3-D Integration. The authors would also like to acknowledge the Core Research Facilities (CRF) at SUSTech for the facilities used, and the technical support provided by the staff and engineers at the CRF.

## Author contributions

G.Z. and F.Z. jointly conceived the project. F.Z. and G.Z. wrote the manuscript. G.Z. conducted the MSFP synthesis, electrical and optical measurements, and memory crossbar array fabrication. J.L., X.H., and F.Z. constructed the memory array control system and algorithm. G.Z., L.C., X.C., W.W., B.S., G.X., and Z.R. conducted the SFP synthesis. Q.S. provided important guidance and experimental instrument support. S.D., Q.S., and L.W. provided platform support.

## Competing interests

The authors declare no competing interests.
