## [Peer Review File · Nature Communications]

REVIEWER COMMENTS

Reviewer #1 (Remarks to the Author):

Manuscript Title: Full Hardware Implementation of Neuromorphic Visual System Based on Multimodal Optoelectronic Resistive Memory Arrays for Versatile Image Processing (NCOMMS-23-25136-T)

This paper suggested that resistive memory arrays for versatile image processing, which was inspired by visual system. However, authors should revise the manuscript as following comments.

Comments:

1. Authors claimed that modified silk fibroin protein (MSFP) is more effective, however, authors have not compared it with unmodified silk fibroin protein (ISFP). Author should give comparative results between MSFP and UMSFP at least with respect to the IV curves.
2. Further, to support the benefits of modification of silk fibroin protein, authors should provide the change in crystallographic and structural properties of UMSFP to MSFP with the help of the following characterization techniques such as XRD, and SEM.
3. In this work, authors have used Au as a top and bottom electrode, however, we suggest authors should check their performance with several other electrode materials like Al, Ag, and ITO.
4. In order to argue with the current technology, authors should give a comparison table of your material efficiency concerning the previously reported materials.
5. Authors should explain, what are the capabilities of the ORRAM mode array in image pre-processing, and what functions can the ERRAM mode array perform?
6. With background denoising techniques from the images, most important features can be removed. How to define that important features are kept after applying background denoising? Authors should define them.
7. In Figure 3, authors show NRM and PPM effects. Especially, Figure 3d and e also show stability. However, Figure 3e is a little bit inconsistent in all respects compared to Figure d. Hence, authors should clearly explain NPM intensity flow and stability more.
8. How does the trap concentration in the MSFP-based memory affect its resistive switching behavior? Authors should describe it.
9. The computation complexity of the proposed method should be clearly described.

10. In Figure 3g, how much epoch or repeat to get this result? Authors should define it.
11. Authors should give a more detailed explanation of the computer simulation for Figure 4a.
12. Authors should put the resulting number in Figure 4g and h.
13. In Figure 4i, it is hard to understand, so authors should explain the figure in more detail.

Reviewer #2 (Remarks to the Author):

The article by Zhou, et al discusses testing and implementation of silk-protein based memristors that exhibit multi-modal variable conductance in response to both light exposure and applied voltage. In addition to voltage-controlled resistive switching, they show that device conductance is enhanced upon exposure to 405nm light at intensities above 70mW, but it is lowered at intensities between 10-60mW (with minimum conductance occurring at 60 mW). This bidirectional change in baseline conductance enables them to perform contrast enhancement and denoising of images through a seemingly intrinsic intensity threshold to positive and negative changes in conductance. Because the same memristor can also be electrically programmed via resistive switching, they demonstrate in hardware that two memristor arrays can be built to perform image preprocessing and convolutional neural network-based classification. The device properties are interesting and the application of their use is potentially compelling. However, there are several key weaknesses in this manuscript that prevent it from being acceptable for publication. The reviewer suggests a major revision be considered to allow them a chance to fix these issues.

Major

1. Missing device details: what is the thickness, footprint, and total area of each device?
2. Electrical resistive switching: What sweep rate(s) were used for the i-v traces shown in Figure 2c? how quickly do the devices change conductance? Is the memory volatile or nonvolatile? The counterclockwise hysteresis paths in both quadrants 1 and 3 suggest the devices exhibit volatile memory. But the data in Figure S3c show nonvolatile states were written (at unknown voltage levels, durations—details needed here)! Therefore, the reviewer questions whether these devices exhibit volatile or nonvolatile memory resistance. Their characteristic speeds of switching and memory loss also aren't disclosed. For context, what dialyzing time/conditions were used to create the MSFP for the devices described in Figure 2? It isn't clear from Figure 2d that changes are representative of LTP or LTD (i.e. do these changes persist after writing?)

3. The paper makes little mention of device repeatability/stability and no mention of device-to-device reproducibility. These are major gaps that limits understanding how “identical” the various memristors are in the CNN arrays and how many times they can be programmed.
4. The proposed switching mechanism isn’t backed by specific data other than i-v slope fitting. The authors state “The analogue resistive memory behaviours with multiple states are dependent on the ion doping concentration of Br⁻ and Li⁺ in the MSFP thin film.” But this isn’t supported by any data. Therefore, the reviewer requests that the mechanism be described as “proposed” or “possible,” and that more evidence be used to support their statements.
5. Test conditions and data presentation for ORRAM: What was the light exposure duration/distance for the data in Figure 2f-k, and what was the ERRAM state (HRS or LRS)? How long does optical switching take? What is the switching repeatability and device-to-device reproducibility in these ORRAM responses? And do you see similar trends with cs-AFM across different regions of a single device? What caused the extra current spikes shown in 3c (i.e., 2 light pulses, but 3-4 current pulses?) It is unclear how plots on right of 3d and 3e correspond to plots on left (i.e. when were these long measurements recorded relative to the incremental increase in light intensity?)
6. How does electrical switching impact optical switching in the same device? If a device is switched from HRS to LRS using +0.7-1V, does this change the baseline conductance set by light? What happens if it’s then switched back to HRS using voltage? Does the same baseline restore? Or does it erase light-induced programming?
7. The authors use FTIR to investigate changes in protein secondary structure upon exposure to light. But it remains unclear why 60 mW is a local min/max for Bs and Bt? How reproducible is this result? How does the thickness/area of the sample effect this critical intensity and changes in conductance upon light exposure?
8. Image processing in Figure 3: how are images pre-configured and “mapped” onto an array of 12x12 devices? Duration, pulse #? What properties do these images have (e.g., image type/requirements? color/binary? of varying intensities at 1 wavelength)?
9. “After completing the image preprocessing, the MSFP-based memristor crossbar array could be erased by the light through the NPM effect with zero electrical power consumption.” But was power supplied to the light that you aren’t counting?
10. Are the two banks of MSFP memristors really identical? Identical in composition, #, and dimensions?
11. The configuration/operation of the neural network layers is hard to follow: Where is the fully connected network? “Considering the influence of interconnect resistance, the first three row of the memristor array in the rear 12x12 ERRAM mode array are not operated.” What does this mean? Consider adding/improving/labeling additional image(s) to show how information flows through the system. Also, the 3x3 convolution kernel and its use is unclear. Also, what determines the size of this kernel? # of images? number of pixels/devices? images aren't 3x3. “The convolution kernel adopted here for feature extraction is $[[1/9,1/9,1/9], [1/9,1/9,1/9], [1/9,1/9,1/9]]$ ” How were these values selected/determined?

12. Simulations used in Figure S9 and S10 are not described anywhere.

13. Power consumption or overall device footprints are not discussed despite suggesting this approach would lead to more compact, efficient systems for image processing.

Minor

- pg 4. “shortening” not “shorting”
- The caption for Figure 1 does not comprehensively explain all parts of the figure. And these are not adequately discussed in the text. Either amend the caption or remove undiscussed content.
- Figure 2b fails to label Pg3 and DHI compounds. And it isn’t clear what Sericin is.
- The use of LRS and HRS terms are confusing for describing the changes in device conductance upon exposure to 405nm light at different intensities. maybe use different labels to differentiate from the ERRAM switching LRS/HRS states.
- The FTIR spectrum for the MSFP film before light illumination shown in Figure S5 should be plotted in same wavenumber range and y-scale as Figures 2g-h for direct comparison.
- Figure 2i isn’t mentioned in the text.
- Poor/confusing wording throughout:
 - o “...secondary structure outweighs of the decreased β -t (5.77%) secondary structure leading to the PPM effect.” don't both of these changes drive conductivity increase? therefore, one isn't outweighing the other to cause PPM.
 - o
 - o “The increasing amount β -t secondary structures then becomes higher than that of β -t secondary structures, which leads to an NPM effect.”
 - o “The PPM effect presents an intensified growth trend with the pulse number under a higher light intensity.” Intensified growth = nonlinear increase in conductance?
- Pg 15, “after 50 pulse stimulation”. after 200 pulses?
- Define all acronyms: ADC, DAC, TIA, ARM,
- Conclusion: “high image recognition accuracy of 95%.” Do you mean 99.5%?

Point-by-point Response

We are grateful for the reviewers' positive comments on our work. We also thank for the reviewers' time and efforts in providing us with detailed and valuable comments. These suggestions are greatly helpful to us in improving the quality of this work. We have addressed all the concerns raised by the reviewer point-by-point as follows and highlighted the revised parts in yellow.

Reviewer #1

Full Hardware Implementation of Neuromorphic Visual System Based on Multimodal Optoelectronic Resistive Memory Arrays for Versatile Image Processing (NCOMMS-23-25136-T). This paper suggested that resistive memory arrays for versatile image processing, which was inspired by visual system. However, authors should revise the manuscript as following comments.

Response: Thanks very much for your constructive comments and suggestions on this work. All the comments and suggestions are very precious to us to further improve the quality of this work.

Comment 1:

Authors claimed that modified silk fibroin protein (MSFP) is more effective, however, authors have not compared it with unmodified silk fibroin protein (USFP). Author should give comparative results between MSFP and UMSFP at least with respect to the $I-V$ curves.

Response: Thanks very much for the valuable suggestion. The resistive switching memory behaviour exhibits obvious differences between the unmodified silk fibroin protein (USFP) memory and the modified silk fibroin protein (MSFP) memory. The USFP memory exhibits a bipolar resistive switching memory behaviour with sharp SET and RESET processes, demonstrating digital switching features, while the MSFP memory shows a bipolar resistive switching memory behaviour with continuous SET

and RESET processes, demonstrating analogue switching features.

Figures R1a-b are the typical I - V curves of the USFP memory with the precursor dialyzing times of 48 and 36 hours, respectively, all demonstrating digital-type switching featured by abrupt SET and RESET process which may be attributed to the formation of the Li^+ -based conduction paths. By comparison, the I - V curves of the MSFP memory device with different precursor dialyzing times of 48 and 36 hours exhibit analogue switching memory behaviours which may be attributed to the space charge limited current (SCLC) mechanism since most Li ions can be pinned by the rich hydrogen bond formed between USFP, Pg-3, and 5,6-DHI (Figs. R1c-d), therefore the electron trapping and de-trapping becomes the dominating mechanism.

To be more specific, the differences in memory switching between the USFP memory device and MSFP memory device may be attributed to the strong hydrogen-bond formation in the MSFP film. In the USFP memory device, without the hydrogen-bond effect, the USFP network structure is relatively sparser, therefore the Li ions are relatively mobile in the USFP thin film to form Li-ion based conduction path. While due to the formation of strong hydrogen bonds in the MSFP thin film as shown in Figure R2, the protein network in the MSFP film is relatively denser, leading to much lower Li ion mobility in the MSFP thin film. In addition, since the residual ion such as Li^+ can introduce a certain number of trap sites in the MSFP layers¹⁻³, the electron trapping and de-trapping in the MSFP thin film may become the dominating mechanism, contributing to the continuous SET and RESET process (analogue switching). Therefore, considering the multilevel storage capability for in-memory computing, the MSFP devices are adopted for systematic investigation in the manuscript. The revision has been added to the revised supplementary information.

Fig. R1. Comparisons of the resistive switching behaviours between (a)-(b) USFP- and (c)-(d) MSFP-based memory devices with different precursor dialyzing times.

Fig. R2. FTIR spectra of the USFP and MSFP thin film. The major structures of USFP are reserved in the MSFP thin film while the $-\text{CH}_2-$ -based chains emerge in the MSFP sample.

References

1. Wang, W., et al. An analogue memristor made of silk fibroin polymer. *J. Mater. Chem. C* **9**, 14583-14588 (2021).
2. Min, K., Umar, M., Ryu, S., Lee, S., Kim, S. Silk protein as a new optically transparent adhesion layer for an ultra-smooth sub-10 nm gold layer. *Nanotechnology* **28**, 115201 (2017).
3. Shi, C., et al. New silk road: from mesoscopic reconstruction/functionalization to flexible meso-electronics/phonics based on cocoon silk materials. *Adv. Mater.* **33**, 2005910 (2021).

The corresponding revision is as follows:

“In comparison with the SFP-based memory showing digital switching and abrupt set and reset process (Supplementary Figures 8a-c), the Au/MSFP/Au memory device shows an analogue switching behaviour with nonvolatile 16 multilevel resistance states as shown in Supplementary Figure 9.”

“To further examine the structural change from SFP to MSFP thin films, the Fourier transform infrared spectroscopy (FTIR) characterizations are conducted, which suggest new structures such as -CH₂-based chains formed and no observation of the chemical bond of C-O located at 1165 cm⁻¹ in the MSFP films (Supplementary Figure 4), indicating that the change of C-O-based bond groups possibly provide the chemical reaction sites to form a large number of hydrogen bonds among SFP, Pg-3 and 5,6-DHI.”

Comment 2:

Further, to support the benefits of modification of silk fibroin protein, authors should provide the change in crystallographic and structural properties of USFP to MSFP with the help of the following characterization techniques such as XRD, and SEM.

Response: Thanks very much for the reviewer’s suggestion. The changes in crystallographic and structural properties of USFP and MSFP were characterized by the XRD and SEM in Figure R3.

Figure R3a exhibits the XRD patterns of the USFP thin films with different precursor dialyzing times of 12, 24, 36, 48 and 72 hours, a mixture of Pg-3 and USFP with the dialyzing time of 48 hours, a mixture of 5,6-DHI and USFP with the dialyzing time of 48 hours, and the MSFP thin film with the precursor dialyzing time of 48 hours. Both USFP and MSFP samples present poor crystallinity. In addition, no obvious crystallographic changes are detected between USFP and MSFP thin films. Then to further examine the structural change, we conducted the Fourier transform infrared

spectroscopy (FTIR) characterizations of the USFP and MSFP thin films as shown in Figure R2, which suggest new structures such as -CH₂-based chains are formed in the MSFP thin film compared with that in the USFP thin film, as shown in Figure R3. The FTIR results of USFP and MSFP demonstrated that the strong chemical bond vibration of the -OH, -C=O, C=C, C-OH respectively located at 3278, 1635, 1520 and 1230 cm⁻¹ existed in both USFP and MSFP thin films while a weak -CH₂(s) vibration located at 2856 cm⁻¹ emerged in the MSFP thin film. It indicates that the major structures of the USFP can be reserved in the MSFP after the chemical reaction between SFP, 5,6-DHI and Pg-3 and the structures such as the connection between protein chains in the MSFP are changed during the reaction. In addition, the chemical bond of C-O located at 1165 cm⁻¹ in the USFP is not detected in the MSFP, indicating that the C-O-based bond groups can possibly provide the chemical reaction sites to form a large number of hydrogen bonds among USFP, Pg-3 and 5,6-DHI, as shown in Figure R2.

Figures R3b-f show the field emission scanning electron microscope (FE-SEM) images of the USFP sample with different precursor dialyzing times from 12 to 72 hours, showing no obvious change in the USFP surface. After reaction with the Pg-3, the 5, 6-DHI, or both of them, the formed MSFP film surface becomes denser and smoother, showing higher thin film quality (Figs. R3g-i), which may be responsible for the lower current density and higher stability in the MSFP memory device compared with that in the USFP memory device. The revision has been added to the revised supplementary information.

Fig. R3. Crystallographic and structural characterizations of USFP and MSFP thin films. **a**, The XRD pattern of the USFP thin film with various precursor dialyzing times from 12 to 72 hours, and the MSFP thin film formed by the reaction of Pg-3, 5,6 DHI with USFP. **b-f**, FE-SEM images of the USFP thin films with dialyzing times of 12, 24, 36, 48, and 72 hours, respectively. **g-j**, FE-SEM image of the USFP thin films and MSFP thin film (dialyzing time of 48 hours), respectively. The scale bar represents 500nm.

The corresponding revision is as follows:

“The MSFP thin film shows a denser and smoother surface than the SFP thin film, showing higher thin film quality (Supplementary Figure 3). To further examine the structural change from SFP to MSFP thin films, the Fourier transform infrared spectroscopy (FTIR) characterizations are conducted, which suggest new structures such as -CH₂-based chains formed and no observation of the chemical bond of C-O located at 1165 cm⁻¹ in the MSFP films (Supplementary Figure 4), indicating that the

change of C-O-based bond groups possibly provide the chemical reaction sites to form a large number of hydrogen bonds among SFP, Pg-3 and 5,6-DHI.”

Comment 3:

In this work, authors have used Au as a top and bottom electrode, however, we suggest authors should check their performance with several other electrode materials like Al, Ag, and ITO.

Response: Thanks very much for the suggestion. We further compare the switching behaviours in the MSFP memory devices with different top electrodes such as Au, Al, Ag, and ITO. Figure R4 shows the I - V switching in the Au/MSFP/Au, Al/MSFP/Au, Ag/MSFP/Au and ITO/MSFP/Au memory devices, respectively. All devices exhibit similar analogue switching memory behaviours, indicating that this resistance switching mainly relies on electron trapping/detrapping in the MSFP trapping sites.

For the Au/MSFP/Au memory device, the memory behaviours are featured by anticlockwise I - V sweepings, analogue switching, and a resistance ratio of ~ 10 (Figure R4a). For the Ag/MSFP/Au and Al/MSFP/Au memories, both of the memory devices exhibit similar switching behaviours with the Au/MSFP/Au memory device (Figures R4b-c). Generally, the Ag electrode and Al electrode can be electric-field-driven to be Ag ions and Al ions respectively and the migration in the switching function layer to form filaments that would exhibit an abrupt switching behaviour. However, the similar continuous switching observations for the Ag/MSFP/Au and Al/MSFP/Au memory devices compared with the Au/MSFP/Au device indicate that these resistive switchings are mainly dominated by the proposed electron trapping/detrapping mechanism in the MSFP function layer. This may be due to the dense network formed in the MSFP thin film, which may prevent the migration of Ag and Al ions to a certain degree. The ITO/MSFP/Au memory device also shows a similar analogue resistive switching compared with the Au/MSFP/Au device (Figure R4d). In comparison, the MSFP

memory devices with different top electrodes such as Au, Ag, Al and ITO electrodes show similar analogue switching memory behaviours, resistance ratio and current magnitude. Therefore, the electron trapping and de-trapping in the MSFP switching function layer may play a dominant role in resistive switching.

Fig. R4. Comparisons of the switching behaviours in the MSFP memory devices with different top electrodes such as (a) Au, (b) Al, (c) Ag, and (d) ITO.

Comment 4:

In order to argue with the current technology, authors should give a comparison table of your material efficiency concerning the previously reported materials.

Response: Thanks for the constructive suggestion.

The superior advantages of our memory device are the multimodal resistive switching including optical multilevel resistive switching (ORRAM mode) and electrical multilevel resistive switching (ERRAM mode) due to the unique properties in the MSFP material. More importantly, the ORRAM mode shows unique positive photoconductance memory (PPM) and negative photoconductance memory (NPM), suggesting the device can be fully optical SET and RESET without the triggering of the electrical stimuli, which can be rarely realized in the previous report. It is the first time to report such a multifunctional and multimodal memory device integrated with both

NPM and PPM optical resistive switching and analogue electrical switching, leveraging from the unique properties in the MSFP materials. Such unique multimodal switching characteristics in the device enable it to mimic photoreceptor cells, bipolar cells, and visual cortex, and build a neuromorphic visual system for versatile low-level and high-level processing functions with only one type of memory array, which has also not been realized in the previous report. In addition, the Au/MSFP/Au device is free-standing, flexible, and environmentally friendly, showing great potential in future flexible electronics. For the optical memory switching, the device shows good nonvolatile and multi-level resistive switching with a response time of ~ 100 ms, retention time of $\sim 10^4$ s, and low energy consumption of ~ 5 pJ. For the electrical memory switching, our MSFP memory also shows good nonvolatile and multi-level resistive switching with low voltage operation of 1 V and low power consumption of ~ 250 pJ. We compared the optoelectronic memory devices based on different photoelectronic function materials (including oxides, 2D materials, and polymers) with our material, as shown in Table. R1. The revision has been added to the revised manuscript and supplementary information is as follows:

“The comparison of our device in both optical switching and electrical switching with the devices based on other materials is shown in Supplementary Table 4.”

Table R1. Comparison of the optoelectronic memory devices based on different photoelectronic function materials.

Device	Material	Optical switching memory	Fully optical operation	Optical response	Optical retention	Electrical switching memory	Electrical switching type	Electrical retention	Electrical operation voltage	Energy Consumption	On/off ratio	Multimodal	Applications	Full hardware system
Memristor (Ref.1)	MoO _x	Yes, opt. memory	No	100ms	>10 ² s	No	No	-	2.5V/0.1V	-	-	No	contrast enhancement; denoising	No
Memristor + diode (Ref.2)	InGaAs-based	No, opt. response	No	100ms	volatile	Yes	digital	>10 ⁴ s	~5V/~1V	-	~10 ²	Yes	image classification	No
Memristor (Ref. 3)	GeSe ₃	Yes, opt.	No	5s	~10 ² s	Yes	digital	>10 ⁴ s	~0.6/-	0.39mJ	~10 ³	No	logical operation	No
Memristor (Ref. 4)	Graphene	No, opt.	No	2s	volatility	Yes	digital	>10 ⁴ s	~6V/0V	-	~10 ³	No	image classification	No
Memristor (Ref. 5)	InGaZnO	Yes, opt. memory	Yes	~2s	~10 ⁴ s	Yes	analogue	>10 ⁴ s	~2.0V/10mV	-	~5	No	-	No
Transistor (Refs. 6-9)	WSe ₂ -based	Yes, opt. memory	No	~200ms	~10 ² s	-	No	-	1.5~9V/-	-	-	No	motion detection;	No
Transistor (Refs.10-16)	MoS ₂ -based	No, opt. response	No	~5ms	volatile	-	-No	-	3~6V/-	-	-	No	perception encoding;	No
Transistor (Refs. 17-22)	Organic-inorganic-based	No, opt. response	No	~5ms	volatile	-	-No	-	~40V/	-	-	No	-	No
Memristor (Our work)	modified silk fibroin protein	Yes, opt. memory	Yes	100ms	>10 ⁴ s	Yes	analogue	>10 ⁴ s	1V/0.1V	~250pJ	~10	Yes	feature extraction; contrast enhancement; background denoising; image recognition	Yes

References

1. Zhou, F., et al. Optoelectronic resistive random-access memory for neuromorphic vision sensors. *Nat. Nanotechnol.* **14**, 776-782 (2019).
2. Lee, D., et al. In-sensor image memorization and encoding via optical neurons for bio-stimulus domain reduction towards visual cognitive processing. *Nat. Commun.* **13**, 5223 (2022).
3. Syed, S., et al. Chalcogenide optomemristors for multi-factor neuromorphic computation. *Nat. Commun.* **13**, 2247 (2022).
4. Fu, X., et al. Graphene/MoS_{2-x}O_x/graphene photomemristor with tunable non-volatile responsivities for neuromorphic vision processing. *Light Sci. Appl.* **12**, 39 (2023).
5. Hu, L., et al. All-optically controlled memristor for optoelectronic neuromorphic computing. *Adv. Funct. Mater.* **31**, 2005582 (2021).
6. Mennel, L., et al. Ultrafast machine vision with 2D material neural network image sensors. *Nature* **579**, 62-66 (2020).
7. Chen, H., Xue, X., Liu, C., Fang, J., Wang, Z., Wang, J., & Zhou, P. Logic gates based on neuristors made from two-dimensional materials. *Nat. Electron.* **4**, 399-404 (2021).
8. Wang, C. Y. et al. Gate-tunable van der Waals heterostructure for reconfigurable neural network vision sensor. *Sci. Adv.* **6**, eaba6173 (2020).
9. Zhang, Z., et al. All-in-one two-dimensional retinomorphic hardware device for motion detection and recognition. *Nat. Nanotechnol.* **17**, 27-32 (2022).
10. Chen, J., Chai, Y., et al. Optoelectronic graded neurons for bioinspired in-sensor motion perception. *Nat. Nanotechnol.* doi:10.1038/s41565-023-01379-2 (2023).
11. Liao, F., Chai, Y., et al. Bioinspired in-sensor visual adaptation for accurate perception. *Nat. Electron.* **5**, 84-91 (2022).
12. Yu, J., et al. Bioinspired mechano-photonic artificial synapse based on graphene /MoS₂ heterostructure. *Sci. Adv.* **7**, eabd9117 (2021).
13. Syed, S., et al. Atomically thin optomemristive feedback neurons. *Nat. Nanotechnol.* doi: 10.1038/ s41565-023-01391-6 (2023).
14. Wang, F., et al. A two-dimensional mid-infrared optoelectronic retina enabling simultaneous perception and encoding. *Nat. Commun.* **14**, 1938 (2023).
15. Dodda, A., et al. Active pixel sensor matrix based on monolayer MoS₂ phototransistor array. *Nat. Mater.* **21**, 1379-1387 (2022).
16. Ma, S., et al. A 619-pixel machine vision enhancement chip based on two-

- dimensional semiconductors. *Sci. Adv.* **8**, eabn9328 (2022).
17. Chen, K., et al. Organic optoelectronic synapse based on photon-modulated electrochemical doping. *Nat. Photon.* **17**, 629-637 (2023).
 18. Zhu, Q., et al. A flexible ultrasensitive optoelectronic sensor array for neuromorphic vision system. *Nat. Commun.* **12**, 1798 (2021).
 19. Jiang, T., et al. Tetrachromatic vision-inspired neuromorphic sensors with ultraweak ultraviolet detection. *Nat. Commun.* **14**, 2281 (2023).
 20. Lee, S., et al. Programmable black phosphorus image sensor for broadband optoelectronic edge computing. *Nat. Commun.* **13**, 1485 (2022).
 21. Xu, Y., et al. Optically readable organic electrochemical synaptic transistors for neuromorphic photonic image processing. *Nano Lett.* **23**, 5264-5271 (2023).
 22. He, S., et al. Reconfigurable optoelectronic synaptic transistor with stable Zr-CsPbI₃ nanocrystals for visuomorphic computing, *Adv. Mater.* **35**, 2208479 (2023).

Comment 5:

Authors should explain, what are the capabilities of the ORRAM mode array in image pre-processing, and what functions can the ERRAM mode array perform?

Response: Thanks very much for the suggestion.

Figures R5a-b (Figures 4c-d) show the whole CNN neural network structure implemented based on the ORRAM mode array and ERRAM mode array. The ORRAM mode array can not only *in-situ* complete image preprocessing functions such as contrast enhancement, and background denoising but also perform optical convolutional operation for image feature extraction. The ERRAM mode array can complete the convolution layer computations and fully connected layer computations in the convolutional neural networks (CNN). Assisted by the external peripheral circuit, the whole CNN can be completed for image recognition.

More specifically, for the ORRAM mode array, the contrast enhancement and background denoising are performed according to the NPM and PPM operations in the ORRAM mode array (Figure R5a). Utilizing PPM and NPM effects, the body pattern pixels in the input image represented by relatively higher brightness are enhanced by

the PPM while the background pixels with relatively lower brightness are suppressed by the NPM effect, Therefore, a contrast-enhanced image can be obtained (Figure R5d). Additionally, the ORRAM array can also implement the in-sensor convolutional operation for image feature extraction (Figure R5c). ORRAM can firstly sense and store the image information in the conductances represented using G_{ml} . For the optical convolution process, a reconfigurable 3×3 convolution kernel is first mapped into three reading voltages (V_{11}, V_{12}, V_{13}) at the temporal domain (T_1, T_2, T_3), which is then input into the ORRAM mode memory array for convolutional operations with the stored conductances. The output currents for every column at T_1, T_2 and T_3 are defined to be $I_{11}, I_{12}, I_{13}; I_{21}, I_{22}, I_{23};$ and I_{31}, I_{32}, I_{33} , respectively. The photoductance (G_{ml}), reading voltage (V_{km}), and current (I_{kl}) at each ORRAM array column obey Ohm's law. For each column of the memristor array, the summation current obeys Kirchhoff's law. Therefore, the relationship of cumulative current for each column, read voltage, and conductance can be described as follows:

$$I_{kl} = \sum_{m=1}^3 V_{km} G_{ml} \quad (1)$$

Then new output currents are extracted from a specific current group ($I_{11}, I_{22}, I_{33}; I_{12}, I_{23}, I_{34}; I_{13}, I_{24}, I_{33} \dots$) as follows:

$$I_{out-k,l} = I_{k,l} + I_{k+1,l+1} + I_{k+2,l+2} \quad (2)$$

The summed current $I_{out-k,l}$ represents the result of the convolution operation.

For the ERRAM mode array, the primary function is to perform multiply-accumulate operations for both convolutional operation and fully connected layers in the CNN (Figure R5b). The image feature map output from the ORRAM array is divided into multiple 3×3 patches and input into ERRAM for convolutional computations. The convolution is implemented as follows:

$$I_N = \sum_{m=1}^9 X_m W_m^N = \sum_{m=1}^9 V_m (G_m^{N+} - G_m^{N-}) = I_N^+ - I_N^-$$

Where X_m represents the value of input data, and W_m^N denotes the weight of the

neural network. The weight can be both positive and negative, which is represented using a pair of differential conductances G_m^{N+} and G_m^{N-} . X_m can be mapped to V_m . The V_m and G_m denote the bias voltage applied to the MSFP memory cell and the corresponding conduction value, respectively. I_N is the differential result current between two columns, I_N^+ and I_N^- . The result of the convolution calculation is denoted as I_N . Other network operations, such as max pooling layers and RELU activation function layers, are executed within the ARM core. The visual system based on the ORRAM and ERRAM array shows a high accuracy of 97.3% (Figure R5d). The manuscript has been revised accordingly.

Fig. R5. CNN implementation based on the ORRAM mode array and ERRAM mode array. a, hardware convolution operations in the ORRAM mode array. **b,** hardware convolution operations in the ERRAM mode array for image recognition. **c-d,** image contrast enhancement through PPM and NPM, image feature extraction through hardware convolution operations in the ORRAM mode array. **e,** image recognition results of the system based on the ORRAM and ERRAM arrays.

The corresponding revision is as follows:

“The image preprocessing algorithms and a convolutional neural network (CNN) are

deployed on the hardware system. The 12×12 ORRAM array executes the image preprocessing of contrast enhancement and background denoising and performs the first-layer optical convolutional operation in CNN for image feature extraction. The ERRAM mode array is employed to complete one convolution layer and fully connected layer computations in the CNN. Other network operations, such as max pooling layers and ReLU activation function layers, are executed within the ARM core.

Firstly, for the ORRAM mode array, contrast enhancement and background denoising are performed according to the NPM and PPM operations in the ORRAM mode array. Utilizing PPM and NPM effects, the body pattern pixels in the input image represented by relatively higher brightness are enhanced by the PPM while the background pixels with relatively lower brightness are suppressed by the NPM effect, Therefore, a contrast-enhanced image can be obtained (Figures 4d-e). Additionally, the ORRAM array can also implement the in-sensor convolutional operation for image feature extraction (Figure 4d). A novel optical convolution algorithm deployed on the ORRAM array as shown in Figure 4c is proposed for in-sensor convolution. Each cell in the ORRAM mode array corresponds to one pixel in the input image. The ORRAM mode array can firstly sense and store the image information in the photoconductance represented using G_{ml} (G_{ml} , $m=1, 2, 3 \dots n$; $l=1, 2, 3 \dots n$). For the optical convolution process, a reconfigurable 3×3 convolution kernel is first mapped into three reading voltages (V_{11}, V_{12}, V_{13}) at the temporal domain (T_1, T_2, T_3), which is then input into the ORRAM mode memory array for convolutional operations with the stored photoconductance. The output currents for every column at T_1, T_2 and T_3 are defined to be I_{11}, I_{12}, I_{13} ; I_{21}, I_{22}, I_{23} ; and I_{31}, I_{32}, I_{33} , respectively. The photoductance (G_{ml}), reading voltage (V_{km}), and current (I_{kl}) at each ORRAM array column obey the Ohm's law. For each column of the memory array, the summation current obeys Kirchoff's law. Therefore, the relationship of the summation current for each column, read voltage, and photoconductance can be described as follows:

$$I_{kl} = \sum_{m=1}^3 V_{km} G_{ml} \quad (1)$$

In this approach, a convolution kernel is divided into T_1 , T_2 , and T_3 segments. The current summation at T_1 is first collected and stored in the ARM core, followed by the current summation at T_2 and T_3 . Then new output currents are extracted from a specific current group ($I_{11}, I_{22}, I_{33}; I_{12}, I_{23}, I_{34}; I_{13}, I_{24}, I_{33} \dots$) as follows:

$$I_{out-k,l} = I_{k,l} + I_{k+1,l+1} + I_{k+2,l+2} \quad (2)$$

The employment of the novel optical convolutional operation the ORRAM convolution offers a 75% reduction in time complexity compared to the traditional convolution operation method based on RRAM.”

Comment 6:

With background denoising techniques from the images, most important features can be removed. How to define that important features are kept after applying background denoising? Authors should define them.

Response: Thanks very much for the reviewer’s comment. The input image is composed of two parts, the body pattern pixels and the background noisy pixels. The pixels of the body pattern in the input image have relatively higher brightness (higher intensities), while the pixels of the background noises have relatively lower brightness (lower intensities) (Fig. R7).

Specifically, the input image will first be mapped on the ORRAM mode arrays. Due to the PPM and NPM effects, the output current corresponding to the body pattern pixels with higher brightness will continuously increase because of the PPM effect, while the output current corresponding to the background pixels with relatively lower brightness will continuously decrease because of the NPM effect, therefore resulting in an obvious image contrast enhancement between the body pattern and background noise (Figs. R7a-c). Therefore, the important features are enhanced by correspondingly smoothing the background (Fig. R7d).

After the image contrast enhancement, the in-sensor convolutional computations are performed in the ORRAM mode array for image feature extraction. Due to the convolution process, pixel values are influenced by neighbouring pixels, resulting in a feature map after convolution. Therefore, the feature extraction results (feature map) appear to become blurry but represent the specific features in the input image as shown in (Fig. R7d). The obtained features are further input into subsequent neural network layers for classification. The manuscript has been revised accordingly.

Fig. R7. Image preprocessing in the ORRAM mode array.

The corresponding revision is as follows:

“When the images are input into the array, the body pattern pixels with higher light intensities were enhanced through the PPM effect while the background pixels with lower light intensities were reversely suppressed by the NPM effect, resulting in the enlarged pixel signal ratio between the body pattern pixels and the background pixels. With the increased epochs (increased pulse number), the contrast is continuously enhanced (Supplementary Figure 18), leading to an enhanced contrast of ~33 after 25 epochs. By comparison, the input image shows a light intensity ratio of ~5 between the background pixel and body pattern pixel. Therefore, a ~6 times contrast enhancement is obtained after the image preprocessing based on the ORRAM mode array (Figure 3g). Therefore, the body pattern is enhanced by correspondingly smoothing the

background noise.”

“Utilizing PPM and NPM effects, the body pattern pixels in the input image represented by relatively higher brightness are enhanced by the PPM while the background pixels with relatively lower brightness are suppressed by the NPM effect, Therefore, a contrast-enhanced image with smoothed background noise can be obtained (Figures 4d-e). Additionally, the ORRAM array can also implement the in-sensor convolutional operation for image feature extraction (Figure 4d).”

Comment 7:

In Figure 3, authors show NPM and PPM effects. Especially, Figure 3d and also show stability. However, Figure 3e is a little bit inconsistent in all respects compared to Figure d. Hence, authors should clearly explain NPM intensity flow and stability more.

Response: Thanks very much for the comment. In previous Figure 3d-e, the NPM effect under different light intensities (10-60 mW) was individually displayed during 200 pulse stimulations while the PPM effect was displayed from 0 to 500 pulses. We re-plotted the figures and revised the manuscript to avoid inconsistency according to the suggestions, as shown in Figure R8.

The left of Figure R8a (Figure 3d) shows the PPM effect and conductance potentiation with the increased light pulse number (500 in total) and an increased light intensity ranging from 70 to 100 mW (the pulse number used for each light intensity is 50). The conductance is increased with the continuously applied pulse number before reaching a saturation state at each light intensity. Overall, the conductance is increased with the light intensity level. The right of Figure R8a (Figure 3d) demonstrates the non-volatile retention properties of the memorized states after the removal of the pulse sequence with 50 pulses at each light intensity (70, 75, 80, 82, 85, 87, 89, 90, 95, 100 mW). The discrete photoconductance states can be well maintained after turning off the light stimuli, demonstrating the non-volatile multiple photoconductance states. The left

of Figure R8b (Figure 3e) presents the NPM effect and conductance depression with the increased pulse number (1000 in total) at the different light intensities ranging from 10 to 60 mW (the pulse number for each light intensity is 200 since NPM effect is less easily to reach a saturation at each light intensity compared with the PPM effect). The device current presents a continuously decreasing trend with the increased pulse number and the light intensity. The right of Figure R8b (Figure 3e) shows the retention of the conductance state after the removal of the pulse sequence with 200 pulses at each light intensity (10, 15, 25, 40 and 60 mW). After the removal of the light stimuli, the multiple conductance states can be well maintained. The manuscript has been revised accordingly.

Fig. R8. PPM and NPM effects. Left, photocurrent density versus light pulse number at different light intensities ranging from 10 to 100mW; Right, retention time after the removal of the light stimuli with different intensities. **a**, The PPM effect with LTP features under the increased pulse number and the varied light intensities ranging from 70 100 mW. **b**, The NPM effect with LTP features under the increased pulse number and the varied light intensities ranging from 10 to 60 mW.

The corresponding revision is as follows:

“The left of Figure 3d shows the PPM effect and conductance potentiation with the increased light pulse number and an increased light intensity ranging from 70 to 100 mW (pulse number for each light intensity is 50). The PPM effect presents a growth trend with the pulse number under a higher light intensity. Overall, the conductance is increased with the light intensity level. The right of Figure 3d demonstrates the non-volatile retention properties of the memorized states after the removal of the pulse sequence with 50 pulses at each light intensity (70, 75, 80, 82, 85, 87, 89, 90, 95, 100 mW). The discrete photoconductance states can be well maintained after turning off the light stimuli, demonstrating the non-volatile multiple photoconductance states. The left of Figure 3e presents the NPM effect and conductance depression with the increased pulse number at the different light intensities ranging from 10 to 60 mW (pulse number for each light intensity is 200). The device current presents a continuously decreasing trend with the increased pulse number and the light intensity. The right of Figure 3e shows the retention of the conductance state after the removal of the pulse sequence with 200 pulses at each light intensity (10, 15, 25, 40 and 60 mW). After the removal of the light stimuli, the multiple conductance states can be well maintained. The effect of the electrical switching on the optical switching is also investigated, as shown in Supplementary Figure 16. The comparison of our device in both optical switching and electrical switching with the devices based on other materials is shown Supplementary Table 4.”

Comment 8:

How does the trap concentration in the MSFP-based memory affect its resistive switching behavior? Authors should describe it.

Response: Thanks very much for the comment. A suitable trap concentration can lead to the stable and analogue resistive switching memory behaviour.

Since the residual ion such as Li^+ can introduce a certain number of trap sites in the function layers ¹⁻³, we fabricated the MSFP memories with different Li^+

concentrations and thus different trapping concentrations. The Li^+ concentrations can be modulated through the precursor dialyzing times. Therefore, to further examine the effect of trap concentration on the resistive switching behaviour, we prepared the MSFP memory devices with different precursor dialyzing times of 36, 48 and 72 hours respectively and estimate the number of Li^+ in the MSFP materials by the inductively coupled plasma (ICP) with mass spectrometry, as shown in Table R2. The Li^+ concentration in the MSFP materials decreases from 65.3 to 56.08 to 48.74 mg/Kg when the dialyzing time increases from 36 to 48 to 72 hours.

Figures R9a-c show the I - V resistive switching curves of the MSFP-based resistive memories with different Li^+ concentrations and the correspondingly different trap concentrations that are obtained by controlling the precursor dialyzing times. The MSFP resistive memory devices with dialyzing times of 36 and 48 hours all exhibit continuous and graded switching behaviours, while the MSFP memory with a dialyzing time of 72 hours exhibits abrupt switching. For the MSFP memory device with a dialyzing time of 72 hours, the device has the least number of Li^+ and trapping sites, which cannot provide sufficient sites for continuous electron trapping. In comparison, the devices with dialyzing times of 36 hours and 48 hours have more trapping sites than those of 72 hours, therefore exhibiting continuous and graded switching.

Comparing the switching between the devices with the dialyzing times of 36 hours and 48 hours, the device with the dialyzing time of 36 hours shows higher current density and a relatively unstable switching effect, while the device with the precursor dialyzing time of 48 hours shows lower current density and better analogue switching behaviours. This may be attributed to that a longer dialyzing time induces less Li^+ and fewer trap states inside the MSFP thin films. Because of the lower trap concentration, the MSFP memory with a 48-hour precursor dialyzing time shows a lower current density at HRS, a smaller on/off ratio (~ 4 for 5 voltage sweeps) and more stable switching than that with a 36-hour precursor dialyzing time. Although a relatively high

Li^+ concentration and trap concentration will lead to a higher on/off ratio (~ 7 for 5 voltage sweeps) in the MSFP device with a dialyzing time of 36 hours, the device shows a higher HRS current density and relatively unstable switching.

In summary, the analogue switching behaviours mainly rely on the electron soft filling in the Li^+ -introduced traps in the MSFP thin film. The device shows a higher on/off ratio when the trap concentration is increased, however, the higher trap concentration could also lead to a relatively unstable switching and higher HRS current density. Therefore, the MSFP memory with dialyzing 48 hours as the optimizing sample was used in this work. The revision has been added to the revised supplementary information.

Fig. R9. Trap concentration influences on the memory switching behaviours. Au/MSFP/Au memory with dialyzing times of (a) 36, (b) 48 and (c) 72, respectively.

Table R2. Li ion concentration in the MSFP material measured by inductively coupled plasma (ICP) with mass spectrometry.

Dialysis time (hours)	Li ion concentration (mg/kg)
36	65.30
48	56.08
72	48.74

The corresponding revision is as follows:

“To further examine the trap concentration-based resistive switching memory behaviours, the MSFP memories with different Li^+ concentrations and thus different trapping concentrations are fabricated to further prove the switching mechanism as

shown in Supplementary Figures 12a-c and Supplementary Table 1.”

References

1. Wang, W., et al. An analogue memristor made of silk fibroin polymer. *J. Mater. Chem. C* **9**, 14583-14588 (2021).
2. Min, K., Umar, M., Ryu, S., Lee, S., Kim, S. Silk protein as a new optically transparent adhesion layer for an ultra-smooth sub-10 nm gold layer. *Nanotechnology* **28**, 115201 (2017).
3. Shi, C., et al. New silk road: from mesoscopic reconstruction/functionalization to flexible meso-electronics/phonics based on cocoon silk materials. *Adv. Mater.* **33**, 2005910 (2021).

Comment 9:

The computation complexity of the proposed method should be clearly described.

Response: Thanks very much for the reviewer’s suggestion. According to this suggestion, the computation complexity of the proposed method in terms of time and space is described as follows:

The novel optical convolution operation based on the ORRAM array as shown in Fig. R10 (Figure 4c) is proposed for in-sensor convolution computing (ICC). Each cell in the ORRAM mode corresponds to one pixel in the input image. The ORRAM arrays sense the pixel intensity (light intensity) and output photoconductance (G_{ml} , $m=1, 2, 3 \dots n$; $l=1, 2, 3 \dots n$). In the meantime, the ORRAM mode memristor array enables to pre-process the images by implementing the optical convolutional algorithm. For the optical convolution process, a reconfigurable 3×3 convolution kernel is first mapped into three reading voltages (V_{11}, V_{12}, V_{13}) at the temporal domain (T_1, T_2, T_3), which is then input into the ORRAM mode memristor array for convolutional operations. The output currents for every column at T_1, T_2 and T_3 are defined to be I_{11}, I_{12}, I_{13} ; I_{21}, I_{22}, I_{23} ; and I_{31}, I_{32}, I_{33} , respectively.

The photoconductance (G_{ml}), reading voltage (V_{km}), and current (I_{kl}) at each column ORRAM mode memory cell obey Ohm's law. For each column of the memory array, the summation current obeys Kirchhoff's law. Therefore, the relationship of cumulative

current for each column, read voltage, and conductance can be described as follows:

$$I_{kl} = \sum_{m=1}^3 V_{km} G_{ml} \quad (1)$$

Then new output currents are extracted from a specific current group ($I_{11}, I_{22}, I_{33}; I_{12}, I_{23}, I_{34}; I_{13}, I_{24}, I_{33} \dots$) as follows:

$$I_{out-k,l} = I_{k,l} + I_{k+1,l+1} + I_{k+2,l+2} \quad (2)$$

In this approach, a convolution kernel is divided into T_1 , T_2 , and T_3 segments. The current summation at T_1 is first collected and stored in the ARM core, followed by the current summation at T_2 and T_3 .

Therefore, in terms of time, if we define the process of applying a voltage pulse to each row of the ORRAM array and collecting all column currents at time T as one operation cycle, then we need 36 cycles to complete the convolution operations for a 12×12 image. However, for traditional ERRAM convolution, the 3×3 pixel units of the image are stretched into a single column, and the 9×1 vector is sequentially sent to an array that stores ERRAM convolution kernels. For a 12×12 image, it requires 144 cycles to complete all the convolution calculations. Therefore, the ORRAM ICC method requires only 25% of the time to complete the convolution operation compared to traditional ERRAM convolution, which also applies when scaling to larger images. Moreover, extra image sensors are required to be integrated with the ERRAM array. The sensor /ERRAM interface can introduce additional time for the completion of the convolution process.

In terms of space, using in-sensor convolution computing, the current values of all columns are cached for each operation. T_1 , T_2 , and T_3 operations require three times the column storage capacity in total. After these three operations, the cached current values can be shifted and summed to obtain the convolution result for one row. Once the result is obtained, the storage space used for caching the column current values at T_2 and T_3 moments can be released. Although this method requires additional storage space, it is a temporary occupation. After completing the overall image convolution, the ORRAM

in-sensor convolution and traditional ERRAM convolution occupy similar storage space for storing the computational results. For a 12×12 resolution image calculation, compared to traditional ERRAM convolution, the ORRAM convolution occupies $117\% \left\{ \frac{(12+2) \times 12}{12 \times 12} \times 100\% = 117\% \right\}$ of the storage space compared to that of the conventional ERRAM convolution. In summary, the ORRAM convolution offers a 75% reduction in time complexity compared to traditional ERRAM convolution, with only a 17% increase in memory consumption in terms of space complexity. The manuscript has been revised accordingly.

Fig. R10. The convolution operation in the ORRAM mode array.

The corresponding revision is following:

“ The employment of the novel optical convolutional operation the ORRAM convolution offers a 75% reduction in time complexity compared to the traditional convolution operation method based on RRAM. ”

Comment 10:

In Figure 3g, how much epoch or repeat to get this result? Authors should define it.

Response: Thanks very much for the valuable suggestion. As shown in Figures R11c-d (Figures 3f-3g), 25 epochs and 100 epochs are employed for the image enhancement and image erasing, respectively. More specifically, 25 PPM pulses

(85mW, 200ms) and 25 NPM pulses (40mW, 200ms) are employed in the experiment to demonstrate the image contrast enhancement based on the ORRAM array. 100 NPM pulses are used for the image erasing based on the ORRAM array.

In fact, the image contrast enhancement is dependent on the epochs. Figure R11a shows the enhanced image contrast *versus* pulse number and the output current density through the PPM effect (corresponding to the body pixels with higher light intensities) and NPM effect (corresponding to the background pixels with lower light intensities) versus pulse number. It suggests that the contrast between the body pixels and background pixels is gradually enhanced as the pulse number increases, which is due to the enlarged output current between the body pixels and background pixels through PPM and NPM effects with the increased pulse number as shown in Figure R11a. As the pulse number increases to 25, the image contrast can be improved by 6 times compared with the original image. Therefore, 25 epochs are adopted in this demonstration. The manuscript has been revised accordingly.

Fig. R11. Method for image contrast enhancement and erasing through PPM and NPM effects. **a**, image contrast enhancement as a function of pulse number (epochs). **b**, a “fish” image was sensed and enhanced in a 12×12 ORRAM mode array. The memorized pre-processed image in the array can be further erased by light illumination.

The corresponding revision is as follows:

“When the images are input into the array, the body pattern pixels with higher light intensities were enhanced through the PPM effect while the background pixels with lower light intensities were reversely suppressed by the NPM effect, resulting in the enlarged pixel signal ratio between the body pattern pixels and the background pixels. With the increased epochs (increased pulse number), the contrast is continuously enhanced (Supplementary Figure 18), leading to an enhanced contrast of ~ 33 after 25 epochs. By comparison, the input image shows a light intensity ratio of ~ 5 between the background pixel and body pattern pixel. Therefore, a ~ 6 times contrast enhancement is obtained after the image preprocessing based on the ORRAM mode array (Figure 3g).”

Comment 11:

Authors should give a more detailed explanation of the computer simulation for Figure 4a.

Response: Thanks very much for the suggestion. Figure R12a (Figure 4a) illustrates the full hardware architecture of our neuromorphic visual system. We conducted the image sensing and processing fully based on the hardware instead of the computer simulation. Figure R12b (Figure 4b) shows the photo of our full hardware system, composed of an ORRAM mode array, an ERRAM mode array, a memory array peripheral control system and an STM32 system. The detailed operation is described as follows:

A complete CNN neural network consists of one layer of ORRAM convolution, max pooling, ReLU activation function, one layer of ERRAM convolution, max pooling, ReLU activation function, and one fully connected layer, as shown in Figure R12c. For the full hardware implementation, the ORRAM array first perceives image information, and then for the in-sensor convolution operation for image feature extraction. The values of the convolution kernels, ranging from -1 to 1, are mapped to a voltage range of -0.2V to 0.2V generated by the DAC and then input into the ORRAM

array. The TIA on the column acts as a virtual ground, converting the incoming current into voltage, which is further collected and converted into a digital signal by the ADC. As depicted in Figure R12c, a 2×2 max-pooling operation and ReLU activation function are performed within the ARM core based on these extracted features. Subsequently, these features are further input into ERRAM for convolution and fully connected layer computations. Due to the ReLU activation function, all feature values become positive. When the features are input into the ERRAM array, all feature values are normalized to the range of 0 to 1, and then mapped to the voltage range of 0 to 0.2V. These normalized feature values are converted into voltage pulses using the DAC and are input into the ERRAM. The convolution operation is performed first, with the TIAs on the columns acting as virtual grounds, collecting the currents on the columns and converting them back into voltages. Afterwards, a 2×2 max-pooling operation and ReLU activation function are executed within the ARM core. Finally, the processed feature values are input into the ERRAM "fully connected layer" to perform classification and output the results. The manuscript has been revised accordingly.

Fig. R12. Fully hardware implementation of the neuromorphic visual system based on the multimodal resistive memory arrays with ORRAM mode array for in-sensor pre-processing and the ERRAM mode array for near-sensor high-level processing. **a**, Architecture of the fully hardware-implemented neuromorphic visual system composed of an ORRAM mode array, an ERRAM mode array, a memory array peripheral control system and an STM32 system. **b**, Optical image of the MSFP memory array-based system. **c**, The whole CNN neural network structure implemented with the ORRAM mode array and ERRAM mode array.

The corresponding revision in the manuscript is as follows:

“The image preprocessing algorithms and a convolutional neural network (CNN) are deployed on the hardware system. The 12×12 ORRAM array executes the image preprocessing of contrast enhancement and background denoising and performs the first-layer optical convolutional operation in CNN for image feature extraction. The ERRAM mode array is employed to complete one convolution layer and fully connected layer computations in the CNN. Other network operations, such as max pooling layers and ReLU activation function layers, are executed within the ARM core.”

“Hardware system workflow

A complete CNN neural network consists of one layer of ORRAM convolution, max pooling, ReLU activation function, one layer of ERRAM convolution, max pooling, ReLU activation function, and one fully connected layer. For the full hardware implementation, the ORRAM array first perceives image information, and then for the in-sensor convolution operation for image feature extraction. The values of the convolution kernels, ranging from -1 to 1, are mapped to a voltage range of -0.2V to 0.2V generated by the DAC and then input into the ORRAM array. The TIA on the column acts as a virtual ground, converting the incoming current into voltage, which is further collected and converted into a digital signal by the ADC. A 2×2 max-pooling

operation and ReLU activation function are performed within the ARM core based on these extracted features. Subsequently, these features are further input into ERRAM for convolution and fully connected layer computations. Due to the ReLU activation function, all feature values become positive. When the features are input into the ERRAM array, all feature values are normalized to the range of 0 to 1, and then mapped to the voltage range of 0 to 0.2V. These normalized feature values are converted into voltage pulses using the DAC and are input into the ERRAM. The convolution operation is performed first, with the TIAs on the columns acting as virtual grounds, collecting the currents on the columns and converting them back into voltages. Afterwards, a 2×2 max-pooling operation and ReLU activation function are executed within the ARM core. Finally, the processed feature values are input into the ERRAM "fully connected layer" to perform classification and output the results.

Comment 12:

Authors should put the resulting number in Figure 4g and h.

Response: Thank you very much for your reminder. We have updated Figures 4g and 4h in the manuscript, as shown below, including the numerical results within the figures. The confusion matrices of the classifications of “bear”, “fish”, and “butterfly” images before and after 50 training epochs are shown in Fig. R13a (Figure 4g) and Fig. R13b (Figure 4h), respectively, yielding an accuracy of 97.3%. The manuscript has been revised accordingly.

Fig. R13. Confusion matrices for the image classification based on the ERRAM mode array before and after 50 training epochs, yielding an accuracy of 97.3%.

The corresponding revision in the manuscript is as follows:

“The confusion matrices of the classifications of “bear”, “fish”, and “butterfly” images before and after 50 training epochs are shown in Figures 4g-h yielding an accuracy of 97.3%.”

Comment 13:

In Figure 4i, it is hard to understand, so authors should explain the figure in more detail.

Response: Thanks very much for this comment and suggestion. Figure R14 (Figure 4i) depicts the conductance distributions (weight maps) of the 12×12 MSFP memory array during the training. After initialization, the 12×12 MSFP memory array is in a completely random state. As the training epochs increase, the accuracy of network recognition also improves. We present the conductance distributions for five different training epochs (network accuracies), as shown in Figure R14 (Figure 4i). This result also suggests the good programming capability of the MSFP memory array.

To be more specific, Figure R14 (Figure 4i) exhibits the conductance change of the 12 × 12 MSFP-based memory crossbar arrays from the initial conductance state to the conductance states at the different training epochs from 11 to 20 (corresponding to different network recognition accuracies from 93.7% to 97.3%). Firstly, random electrical pulses are applied to the memristor array in the top 3 rows × 12 columns and the rest of the memories in the 9 rows × 12 columns are biased with the half-reading voltage strategy by the memory control system, corresponding to the initial

conductance state in the memory array. With the training, one can see that the conductance distribution gradually presents a clear boundary between the top 3 rows \times 12 columns and the rest of 9 rows \times 12 columns as the training epoch (accuracy) increases from 11 (93.7%) to 20 (97.3%), as shown in Figure R14 (Figure 4i). The manuscript has been revised accordingly.

Fig. R14 (Figure 4i). The conductance distributions (weight maps) in the ERRAM array under the initial state and different training epochs.

The corresponding revision in the manuscript is as follows:

“Figure 4i depicts the conductance distributions (weight maps) of the 12 \times 12 MSFP memory array during the training. After initialization, the 12 \times 12 MSFP memory array is in a completely random state. As the training epochs increase, the accuracy of network recognition also improves. The conductance distributions for five different training epochs (network accuracies) are presented as shown in Figure 4i.”

Reviewer #2

The article by Zhou, et al discusses testing and implementation of silk-protein based memristors that exhibit multi-modal variable conductance in response to both light exposure and applied voltage. In addition to voltage-controlled resistive switching, they show that device conductance is enhanced upon exposure to 405 nm light at intensities above 70mW, but it is lowered at intensities between 10-60 mW (with minimum

conductance occurring at 60 mW). This bidirectional change in baseline conductance enables them to perform contrast enhancement and denoising of images through a seemingly intrinsic intensity threshold to positive and negative changes in conductance. Because the same memristor can also be electrically programmed via resistive switching, they demonstrate in hardware that two memristor arrays can be built to perform image preprocessing and convolutional neural network-based classification. The device properties are interesting and the application of their use is potentially compelling.

Response: We sincerely thank for the reviewer's positive comments. All the constructive comments and suggestions are very significant for us to improve this work.

Comment 1:

Missing device details: what is the thickness, footprint, and total area of each device?

Response: Thanks very much for the reminder. The footprint of a single memory cell is $200\mu\text{m} \times 200\mu\text{m}$ (Fig. R15a). The array area of the 12×12 Au/MSFP/Au memory crossbar array is 21.16 mm^2 (Fig. R15b). The thicknesses of the top Au electrode, MSFP switching layer and the bottom Au electrode are ~ 35 , ~ 100 , and 35 nm , respectively. (Fig. R15c). The revision has been added to the revised supplementary information.

Fig. R15. **a**, Optical microscopy image of the Au/MSFP/Au memory cells with an effective area of $200 \mu\text{m} \times 200 \mu\text{m}$ in a single cell. **b**, Optical image of a 12×12 Au/MSFP/Au resistive memory crossbar array. **c**, Field emission scanning electron microscopy (FE-SEM) cross-section image of the Au/MSFP/Au resistive memory. The thicknesses of the Au and MSFP layers are $\sim 35 \text{ nm}$ and $\sim 100 \text{ nm}$, respectively.

The corresponding revision in the manuscript is as follows:

“A two-terminal multimodal MSFP-based resistive memory with a cell area of $200\ \mu\text{m} \times 200\ \mu\text{m}$ and a structure of 35 nm Au/100 nm modified silk fibroin protein (MSFP) switching layer/35 nm Au was fabricated on the MSFP flexible substrate (Figure 2a and Supplementary Figure 2). The optical image of a 12×12 crossbar array with an overall area of $4.6\ \text{mm} \times 4.6\ \text{mm}$ is shown in Supplementary Figures 2a-b.”

Comment 2:

Electrical resistive switching: What sweep rate(s) were used for the i-v traces shown in Figure 2c? how quickly do the devices change conductance? Is the memory volatile or nonvolatile? The counterclockwise hysteresis paths in both quadrants 1 and 3 suggest the devices exhibit volatile memory. But the data in Figure S3c show nonvolatile states were written (at unknown voltage levels, durations—details needed here)! Therefore, the reviewer questions whether these devices exhibit volatile or nonvolatile memory resistance. Their characteristic speeds of switching and memory loss also aren't disclosed. For context, what dialyzing time/conditions were used to create the MSFP for the devices described in Figure 2? It isn't clear from Figure 2d that changes are representative of LTP or LTD (i. e. do these changes persist after writing?).

Response: Thanks very much for the comment. According to this comment and concern, we carefully correct and revise the corresponding description in the electrical switching section.

(1) *What sweep rate(s) were used for the i-v traces shown in Figure 2c?*

Response: Figure R16 (Figure 2c) is the typical I - V sweeping curve ($0 \rightarrow 1 \rightarrow 0 \rightarrow -1 \rightarrow 0\text{V}$) with a voltage scanning rate of 0.2V/s of the Au/MSFP/Au resistive memory. The sweep rate has been added in the Figure.

Fig. R16. Typical I - V switching under continuous 100 positive voltage sweeps from 0 V to 1 V to 0 V and continuous 100 negative voltage sweeps from 0 V to -1 V to 0 V with a voltage scanning rate of 0.2 V/s.

(2) *how quickly do the devices change conductance?*

Response: The response speed of the Au/MSFP/Au memory can reach around $\sim 10 \mu\text{s}$, as shown in Fig. R17. When a single electrical pulse with a voltage amplitude of 0.7 V and pulse width of $50 \mu\text{s}$ is applied to the device, the device conductance quickly changes from a lower conductance state to a higher conductance state. The response time can be extracted as $\sim 10 \mu\text{s}$. The revision has been added to the revised supplementary information.

Fig. R17. Conductance response in the Au/MSFP/Au resistive memory. Current response under the electrical pulse stimuli with amplitude of 0.7 V and pulse width of $50 \mu\text{s}$, indicating a response time of $\sim 10 \mu\text{s}$ for the MSFP-based resistive memory.

The corresponding revision in the manuscript is as follows:

“The response speed of the device is illustrated in Supplementary Figure 7, showing a response speed of $10 \mu\text{s}$.”

(3) *Is the memory volatile or nonvolatile? The counterclockwise hysteresis paths in both*

quadrants 1 and 3 suggest the devices exhibit volatile memory. But the data in Figure S3c show nonvolatile states were written (at unknown voltage levels, durations-details needed here)! Therefore, the reviewer questions whether these devices exhibit volatile or nonvolatile memory resistance. Their characteristic speeds of switching and memory loss also aren't disclosed.

Response: The Au/MSFP/Au resistive memory presents nonvolatile electrical switching as shown in Figure R18.

For the electrical switching, Figure R18 (Figure 2c) and Figure R18a show the I - V sweeping and the retention properties of the Au/MSFP/Au resistive memory. It can be noted that the current gradually increases when the bias voltage sweeps from 0 to 1V (stage 1), and the corresponding low resistance at 1.0V can be maintained when the bias voltage sweeps from 1 to 0V (stage 2), forming an anticlockwise I - V curve in positive voltage region. The continuous positive voltage sweeping drives the continuous conductance increasing from HRS to LRS. This low resistance can be maintained under the voltage sweeping from 0 to -1V (stage 3), then the memory resets to a high resistance state when the voltage reverses its sweeping direction from -1 to 0V (stage 4), causing a clockwise I - V curve in the negative voltage region, as shown in Figure R18a (Figure 2c). Similarly, the continuous negative voltage sweeping drives the continuous conductance decreasing from LRS to HRS. To further demonstrate the nonvolatile conductance state, Figure 18b shows the retention properties of the HRS and LRS programmed by voltage sweeping in Figure 15. The HRS and LRS can all be well maintained for over 10^4 seconds. The multilevel storage of 16 conductance states and retention properties are demonstrated in Figure R19. The 16 conductance states are programmed by consecutive electrical pulse groups, with each group composed of five pairs of a programming pulse (0.7V, 50 μ s) followed by a reading pulse (0.1V, 50 μ s). All 16 conductance states exhibit nonvolatile properties with a retention time of up to 10^3 seconds. The revision has been added to the revised supplementary information.

Fig. R18. Nonvolatile properties. **a**, Typical I - V switching under continuous 100 positive voltage sweeps from 0 V to 1 V to 0 V and continuous 100 negative voltage sweeps from 0 V to -1 V to 0 V with a voltage scanning rate of 0.2 V/s. **b**, Retention time of the LRS and HRS read at 0.1 V for 10^4 seconds.

Fig. R19. 16 conductance states programmed by consecutive electrical pulse groups, with each group composed of five pairs of one programming pulse (0.7V, 50 μ s) and one reading pulse (0.1V, 50 μ s). All 16 conductance states exhibit nonvolatile properties with a retention time of up to 10^3 seconds.

The corresponding revision in the manuscript is as follows:

“The device shows good cyclic endurance and nonvolatile memory with a retention time of over 10^4 seconds (Supplementary Figures 6a-b). The cumulative probability of the low resistance state (LRS) and high resistance state (HRS) for 100 different devices suggests good device-to-device stability under the electrical stimuli (Supplementary Figure 6c).”

“The Au/MSFP/Au memory device shows an analogue switching behaviour with nonvolatile 16 multilevel resistance states as shown in Supplementary Figure 9.”

(4) *For context, what dialyzing time/conditions were used to create the MSFP for the devices described in Figure 2? It isn't clear from Figure 2d that changes are representative of LTP or LTD (i. e. do these changes persist after writing?).*

Response: The dialyzing time used for the MSFP device demonstrated in Figure 2 is 48 hours. The description has been added to the revised manuscript. According to the reviewer's suggestion and concern about the long-term storage effects of the LTP and LTD presented in Figure 2d, we further conducted the retention test after the LTP and LTD processes. Figure R20a (LTP in Figure 2d) exhibits the conductance potentiation with consecutively applied 50 positive pulses (0.7V, 50 μ s) for the long-term potentiation (LTP) process and the conductance retention after the removal of the LTP stimulus. It can be noted the conductance state can be well maintained. Figure R20b (LTD in Figure 2d) presents the LTD process with the consecutively applied 50 negative pulses (-0.7V, 50 μ s). The programmed conductance state of LTD can also be maintained after the removal of the LTD stimulus. The read voltage for LTP and LTD and retention tests is controlled as 0.1 V. Therefore, the conductance states programmed by the LTP and LTP pulses demonstrate good nonvolatility. The revision has been added to the revised supplementary information.

Fig. R20. LTP and LTD and the retention properties of the Au/MSFP/Au resistive memory. a, The conductance programmed by the 50 positive pulses (0.7V, 50µs) for LTP can be well maintained after the removal of the pulse stimulus. **b,** The conductance programmed by 50 negative pulses (-0.7V, 50µs) for LTD can also be well maintained after the removal of the pulse stimulus.

The corresponding revision in the manuscript is as follows:

“Figure 2d exhibits the conductance updates with 50 pulses (0.7V, 50µs) for the long-term potentiation (LTP) process and another 50 pulses (-0.7V, 50µs) for the long-term depression (LTD) process. These programmed conductance state after LTP and LTD processes can be well maintained (Supplementary Figure 10).”

Comment 3:

The paper makes little mention of device repeatability/stability and no mention of device-to-device reproducibility. These are major gaps that limits understanding how “identical” the various memristors are in the CNN arrays and how many times they can be programmed.

Response: Thanks very much for this important comment and concern. The device-to-device stability, and cycle-to-cycle stability in the Au/MSFP/Au resistive memory are demonstrated in Figure R21, indicating that good device-to-device reproducibility, and good cycling endurance. Figure 21a shows the cumulative plot of

the HRS and LRS in 100 MSFP memory devices, showing small device-to-device variation. Figure R21b shows repeated I - V sweeping for 1000 cycles, indicating good endurance and repeatability. The revision has been added to the revised supplementary information.

Fig. R21. Device-to-device reproducibility and cycling endurance in the Au/MSFP/Au memory device. a, Cumulative plot of the HRS and LRS in the 100 MSPF memory devices. **b,** Endurance test for 1000 cycles.

The corresponding revision in the manuscript is as follows:

“The device shows good cyclic endurance and nonvolatile memory with a retention time of over 10^4 seconds (Supplementary Figures 6a-b).”

Comment 4:

The proposed switching mechanism isn't backed by specific data other than i - v slope fitting. The authors state “The analogue resistive memory behaviors with multiple states are dependent on the ion doping concentration of Br^- and Li^+ in the MSFP thin film.” But this isn't supported by any data. Therefore, the reviewer requests that the mechanism be described as “proposed” or “possible,” and that more evidence be used to support their statements.

Response: Thanks very much for the reviewer's comments. We apologize for the inappropriate claim in the previous manuscript. We revised the claim into “The analogue resistive memory behaviours with multiple states are possibly dependent on the ion doping concentration such as Li^+ in the MSFP thin film.”

Since the residual ion such as Li^+ can introduce a certain number of trap sites in

the function layers¹⁻³, we fabricated the MSFP memories with different Li⁺ concentrations and thus different trapping concentrations. The Li⁺ concentrations can be modulated through the precursor dialyzing times. Therefore, to further examine the effect of trap concentration on the resistive switching behaviour, we prepared the MSFP memory devices with different precursor dialyzing times of 36, 48 and 72 hours respectively and estimate the number of Li⁺ in the MSFP materials by the inductively coupled plasma (ICP) with mass spectrometry, as shown in Table R2. The Li⁺ concentration in the MSFP materials decreases from 65.3 to 48.74 mg/Kg when the dialyzing time increases from 36 to 48 to 72 hours.

Figures R9a-c shows the *I-V* resistive switching curves of the MSFP-based resistive memories with different Li⁺ concentrations and the correspondingly different trap concentrations that are obtained through controlling the precursor dialyzing times. The MSFP resistive memory devices with dialyzing times of 36 and 48 hours all exhibit continuous and graded switching behaviours, while the MSFP memory with dialyzing time of 72 hours exhibits abrupt switching. For the MSFP memory device with dialyzing time of 72 hours, the device has the least number of Li⁺ and trapping sites, which cannot provide the sites for continuous electron trapping. In comparison, the devices with dialyzing time of 36 hours and 48 hours have more trapping sites than that of 72 hours, therefore exhibiting the continuous and graded switching.

Comparing the switching between the devices with dialyzing time of 36 hours and 48 hours, the device with the dialyzing time of 36 hours shows higher current density and a relatively unstable switching effect, while the device with the precursor dialyzing time of 48 hours shows lower current density and better analogue switching behaviours. This may be attributed to that a longer dialyzing time induces less Li⁺ and fewer trap states inside the MSFP thin films. Because of the lower trap concentration, the MSFP memory with a 48-hour precursor dialyzing time shows a lower current density at HRS, a smaller on/off ratio (~4 for 5 voltage sweeps) and more stable switching than that

with a 36-hour precursor dialyzing time. Although a relatively high Li^+ concentration and trap concentration will lead to a higher on/off ratio (~ 7 for five voltage sweeps) in the MSFP device with a dialyzing time of 36 hours, the device shows a higher HRS current density and relatively unstable switching.

In summary, the analogue switching behaviours mainly rely on the electron soft filling in the Li^+ -introduced traps in the MSFP thin film. The device shows a higher on/off ratio when the trap concentration is increased, however, the higher trap concentration could also lead to a relatively unstable switching and higher HRS current density. Therefore, the MSFP memory with dialyzing 48 hours as the optimizing sample was used in this work. The revision has been added to the revised supplementary information.

Fig. R9. Trap concentration influences on the memory switching behaviours. Au/MSFP/Au memory with dialyzing times of (a) 36, (b) 48 and (c) 72 hours, respectively.

Table R2. Li ion concentration in the MSFP material measured by inductively coupled plasma (ICP) with mass spectrometry.

Dialysis time (hours)	Li ion concentration (mg/kg)
36	65.30
48	56.08
72	48.74

References

1. Wang, W., et al. An analogue memristor made of silk fibroin polymer. *J. Mater. Chem. C*. **9**, 14583-14588 (2021).
2. Min, K., Umar, M., Ryu, S., Lee, S., Kim, S. Silk protein as a new optically transparent

adhesion layer for an ultra-smooth sub-10 nm gold layer. *Nanotechnology* **28**, 115201 (2017).

3. Shi, C., et al. New silk road: from mesoscopic reconstruction/functionalization to flexible meso-electronics/photronics based on cocoon silk materials. *Adv. Mater.* **33**, 2005910 (2021).

The corresponding revision in the manuscript is as follows:

“To further examine the trap concentration-based resistive switching memory behaviours, the MSFP memories with different Li^+ concentrations and thus different trapping concentrations are fabricated to further prove the switching mechanism as shown in Supplementary Figures 12a-c and Supplementary Table 1.”

Comment 5:

Test conditions and data presentation for ORRAM: What was the light exposure duration/distance for the data in Figure 2f-k, and what was the ERRAM state (HRS or LRS)? How long does optical switching take? What is the switching repeatability and device-to-device reproducibility in these ORRAM responses? And do you see similar trends with CS-AFM across different regions of a single device? What caused the extra current spikes shown in 3c (i.e., 2 light pulses, but 3-4 current pulses?) It is unclear how plots on right of 3d and 3e correspond to plots on left (i. e. when were these long measurements recorded relative to the incremental increase in light intensity?)

Response: Thanks very much for the comment. According to the reviewer’s suggestions, we have explained the given data with more details and supplement more experimental results.

(1) *What was the light exposure duration/distance for the data in Figure 2f-k, and what was the ERRAM state (HRS or LRS)? How long does optical switching take?*

Response: For the *I-V* sweeping measurement in Figure R22a (Figure 2f), the light exposure duration used for each resistance transition is 2 seconds. The exposure distance between the laser and device is a constant distance of 25cm. The initial resistance state (dark state) is at the ERRAM HRS state. When the light illumination with 70~100mW is applied to the device, the conductance increases with the light

intensity showing the PPM effect, corresponding to the transition to a lower resistance state. When the light illumination with 10~60mW is applied to the device, the conductance becomes lower than the initial resistance state (even higher resistance state than the ERRAM HRS state), showing the NPM effect. For the FTIR measurements and CAFM measurements in Figures R22b-f (Figures 2g-k), the light duration used for each light intensity is also 2 seconds. For the optical pulse measurement in Figure 3 in the manuscript, the consecutive light pulses with the pulse width of 100 ms are employed for the optical switching to obtain more resistance states. The manuscript has been revised accordingly.

Fig. R22. The I - V curves, FTIR, and current-sensing AFM measurement. **a**, Tuneable and light intensity dependent NPM and PPM effects, indicating the fully optical set and reset processes. FTIR spectra peak fittings of the MSFP thin film secondary structures in the Amide first region under **b**, 80 mW and **c**, 40 mW UV light, respectively, suggesting the light illumination can alter the MSFP secondary structure. **d**, Schematics of optically induced secondary structure changes leading to PPM and NPM effects. The PPM effect is mainly dominated by the net increasing amount of β -sheet secondary structures, while the NPM effect is dominated by the net increasing amount of β -turn secondary structures. **e**, Measurement setup of current-sensor atomic force microscope (CS-AFM) for the Au/MSFP sample. **f**, CS-AFM images of the Au/MSFP sample under a variety of light intensities. The sample exhibited obvious NPM and PPM effects when illuminated by the 10~65 mW and 80~100 mW 405 nm light for 2 seconds, respectively.

The corresponding revision in the manuscript is as follows:

“**Figure 2k**, CS-AFM images of the Au/MSFP sample under a variety of light intensities. The sample exhibited obvious NPM and PPM effects when illuminated by the 10~65 mW and 80~100 mW 405 nm lasers, respectively. **The light-related measurements of f-h and k are conducted in air ambient with relative humidity of 45% with the light exposure duration of 2.0 seconds.**”

(2) *What is the switching repeatability and device-to-device reproducibility in these ORRAM responses?*

Response: The device-to-device and cycle-to-cycle repeatability in the ORRAM mode have been conducted as shown in Figure R23. The programmed resistance states at the intermediate state, PPM state and NPM state are recorded by a small voltage sweep from 0-0.2 V which will not cause the resistance change for 100 cycles, indicating a small cycle to cycle variation (Figure R23a). Figure R23b shows the cumulative plot of the intermediate state, PPM state and NPM state in 100 MSFP memory devices, suggesting good device-to-device reproducibility for the ORRAM mode. The revision has been added to the revised supplementary information.

Fig. R23. Device-to-device and cycle-to-cycle stability in the MSFP memory device with ORRAM mode operation. a, The cyclically programmed resistance states at the intermediate state, PPM state and NPM state. **b**, The cumulative plot of the intermediate state, PPM state and NPM state in 100 different MSFP memory devices.

The corresponding revision in the manuscript is as follows:

“**This optically PPM and NPN switching presents good cycle-to-cycle and device-to-device stabilities** (Supplementary Figures 13a-b).”

(3) *And do you see similar trends with CS-AFM across different regions of a single device?*

Response: The similar trends of the CS-AFM results can be observed in different regions in a single sample. The different regions of ①, ②, and ③ of Au/MSFP sample are illuminated by the 100 mW light to trigger the PPM effect. The conductance shows similar increasing trends in different regions. After completing the above PPM measurement, the 15 mW light illumination is used to trigger the NPM in regions ①, ②, and ③ (Fig. R24), which also show similar decreasing conductance.

Fig. R24. CS-AFM measurement of the PPM (100mW) and NPM (15mW) effects in the different regions of the Au/MSFP sample.

(4) *What caused the extra current spikes shown in 3c (i.e., 2 light pulses, but 3-4 current pulses?)*

Response: We apologize for the measurement inconsistencies in the previous manuscript, which wrongly indicate the pulse numbers in the previous figure. We have re-measured the short-term synaptic plasticity (STP) using the same condition as shown in Figure R25. Figure R25 shows the NPM STP triggered by paired pulses with a pulse width of 100 ms and different light intensities ranging from 10 to 60 mW. The manuscript figure has been revised accordingly.

Fig. R25. NPM STP triggered by paired pulses with a pulse width of 100ms at different light intensities ranging from 10 to 60 mW.

(5) *It is unclear how plots on right of 3d and 3e correspond to plots on left (i. e. when were these long measurements recorded relative to the incremental increase in light intensity?)*

Response: The left of Figure R8a (Figure 3d) shows the PPM effect and conductance potentiation with the increased light pulse number (500 in total) and an increased light intensity ranging from 70 to 100 mW (the pulse number used for each light intensity is 50). The conductance is increased with the continuously applied pulse number before reaching a saturation state at each light intensity. Overall, the conductance is increased with the light intensity level. The right of Figure R8a (Figure 3d) demonstrates the non-volatile retention properties of the memorized states after the removal of the pulse sequence with 50 pulses at each light intensity (70, 75, 80, 82, 85, 87, 89, 90, 95, 100 mW). The discrete photoconductance states can be well maintained after turning off the light stimuli, demonstrating the non-volatile multiple photoconductance states. The left of Figure R8b (Figure 3e) presents the NPM effect and conductance depression with the increased pulse number (1000 in total) at the different light intensities ranging from 10 to 60 mW (the pulse number for each light intensity is 200 since NPM effect is less easily to reach a saturation at each light intensity compared with the PPM effect). The device current presents a continuously decreasing trend with the increased pulse number and the light intensity. The right of Figure R8b (Figure 3e) shows the retention of the conductance state after the removal of the pulse sequence with 200 pulses at each light intensity (10, 15, 25, 40 and 60 mW).

After the removal of the light stimuli, the multiple conductance states can be well maintained. The manuscript has been revised accordingly.

Fig. R8. PPM and NPM effects. Left, photocurrent density versus light pulse number at different light intensities ranging from 10 to 100mW; Right, retention time after the removal of the light stimuli with different intensities. **a**, The PPM effect with LTP features under the increased pulse number and the varied light intensities ranging from 70 100 mW. **b**, The NPM effect with LTP features under the increased pulse number and the varied light intensities ranging from 10 to 60 mW.

The corresponding revision in the manuscript is as follows:

“The left of Figure 3d shows the PPM effect and conductance potentiation with the increased light pulse number and an increased light intensity ranging from 70 to 100 mW (pulse number for each light intensity is 50). The PPM effect presents a growth trend with the pulse number under a higher light intensity. Overall, the conductance is increased with the light intensity level. The right of Figure 3d demonstrates the non-volatile retention properties of the memorized states after the removal of the pulse sequence with 50 pulses at each light intensity (70, 75, 80, 82, 85, 87, 89, 90, 95, 100 mW). The discrete photoconductance states can be well maintained after turning off the light stimuli, demonstrating the non-volatile multiple photoconductance states. The left

of Figure 3e presents the NPM effect and conductance depression with the increased pulse number at the different light intensities ranging from 10 to 60 mW (pulse number for each light intensity is 200). The device current presents a continuously decreasing trend with the increased pulse number and the light intensity. The right of Figure 3e shows the retention of the conductance state after the removal of the pulse sequence with 200 pulses at each light intensity (10, 15, 25, 40 and 60 mW). After the removal of the light stimuli, the multiple conductance states can be well maintained. The effect of the electrical switching on the optical switching is also investigated, as shown in Supplementary Figure 16. The comparison of our device in both optical switching and electrical switching with the devices based on other materials is shown Supplementary Table 4.”

Comment 6:

How does electrical switching impact optical switching in the same device? If a device is switched from HRS to LRS using +0.7-1V, does this change the baseline conductance set by light? What happens if it's then switched back to HRS using voltage? Does the same baseline restore? Or does it erase light-induced programming?

Response: Thanks very much for this comment.

To investigate the impact of the electrical switching on the optical switching, the Au/MSFP/Au resistive memory is first set to the low resistance state (LRS) by 100 electrical pulses (0.7V, 50 μ s). Then at this LRS triggered by the electrical pulses, we further employ 100 PPM optical pulses (80mW, 200ms) to set the device to a lower LRS, and then this lower LRS can be reset back to the low resistance state (LRS) through 100 NPM optical pulses (40mW, 200ms). The LRS is reset back to its initial HRS baseline through 100 electrical pulses (-0.7V, 50 μ s). After the above operation, the memory device can still be set and reset by the light pulses (set by 50 optical pulses (80mW, 200ms) and then reset by 50 optical pulses (40mW, 200ms)), showing unaffected optical switching behaviours, as shown in Fig. R26. Therefore, the electrical

switching nearly does not impact the optical switching behaviours. The revision has been added to the revised supplementary information.

Fig. R26. The conductance changes with the first 100 electrical pulses (0.7V, 50 μ s) and 100 optical pulses (80mW, 200ms) for the set process and 100 optical pulses (40mW, 200ms) and 100 electrical pulses (-0.7V, 50 μ s) for the reset process, followed with another 50 optical pulses (80mW, 200ms) for optical set and 50 optical pulses (40mW, 200ms) for optical reset.

The corresponding revision in the manuscript is as follows:

“The effect of the electrical switching on the optical switching is also investigated, as shown in Supplementary Figure 16.”

Comment 7:

The authors use FTIR to investigate changes in protein secondary structure upon exposure to light. But it remains unclear why 60 mW is a local min/max for B-s and B-t? How reproducible is this result? How does the thickness/area of the sample effect this critical intensity and changes in conductance upon light exposure?

Response: Thanks very much for the reviewer’s comment. According to the comment, the corresponding experiment has been conducted. The detailed explanation is as follows:

(1) *But it remains unclear why 60 mW is a local min/max for B-s and B-t? How reproducible is this result?*

Response: The protein secondary structures such as alpha-helix (α -h), beta-sheet (β -s), beta-turn (β -t) could be mutually converted under external stimuli such as heating and illuminating radiation¹⁻³. For the MSFP thin films, the 60mW light intensity with a

specific wavelength may provide suitable heat and energy for the α - h unwinding to be the β - s and β - t , resulting in an increase of β - t and a decrease of the β - s secondary structure. The secondary structure conversion is quite reproducible as shown in Fig. R27. Figure R27 illustrates the molar ratio changes of the secondary structures with the light power in the samples fabricated in different batches, all demonstrating similar results. However, we have to admit that the critical value of 60 mW is based on our optical illumination setup. The absolute value can be changed for other optical illumination setups with different illumination distances or areas. However, the critical value is quite reproducible for a certain illumination setup.

Fig. R27. Secondary structure conversion under different light intensities.

Reference

1. Zhang, L., Chen, T., Ban, H. & Liu, L. Hydrogen bonding-assisted thermal conduction in β -sheet crystals of spider silk protein, *Nanoscale* **6**, 7786-7791 (2014).
2. Yun, Y. S., et al, Microporous carbon nanoplates from regenerated silk proteins for supercapacitors, *Adv. Mater.* **25**, 1993-1998 (2013).
3. Min, K., Umar, M., Ryu, S., Lee, S., & Kim, S. Silk protein as a new optically transparent adhesion layer for an ultra-smooth sub-10 nm gold layer. *Nanotechnology* **28**, 115201 (2017).

(2) *How does the thickness/area of the sample effect this critical intensity and changes in conductance upon light exposure?*

Response: First, for the effects of the MSFP thickness on the optical switching behaviours, we studied the optical switching behaviours in the MSFP memory with different MSFP thicknesses of ~ 63 , ~ 97 , and ~ 186 nm, respectively, as shown in Figure R28. The thickness of the MSFP shows impacts on the resistive switching behaviours including current density, resistance ratio, and stability. The memory device with ~ 63

nm MSFP function layer shows a relatively high current density ($8\sim 25\text{Acm}^{-2}$ at 0.2V) and a small resistance ratio (~ 3) between PPM and NPM states (Fig. R28a). A higher resistance ratio (~ 10), and lower current density ($0.35\sim 5\text{Acm}^{-2}$ at 0.2V) are obtained in the MSFP memory with ~ 97 nm MSFP thin film (Fig. R28b), which may be attributed to the higher secondary structure change volume compared with that of 63 nm. However, the higher resistance states and even smaller resistance ratio (~ 2) are observed in the MSFP memory with an MSFP thickness of 186 nm, which may be attributed to that the conversion volume of secondary structure in the MSFP thin film is limited under the light illumination with a certain power, therefore the secondary structure conversion ratio and the resistance change ratio is relatively small when the MSFP thin film thickness is relatively thick (186 nm) and the as-prepared MSFP memory device is already at very HRS (Fig. R28c).

For the effects of the device area on the optical switching behaviours, the optical switching of MSFP memory devices with different device areas of $100\times 100\mu\text{m}^2$, $150\times 150\mu\text{m}^2$ and $200\times 200\mu\text{m}^2$ are studied as shown in Figure R28d. It can be noted that the memory device with a large area exhibits a wider conductance range for both the NPM and PPM effects. The memory cell with an area of $200\times 200\mu\text{m}^2$ shows an optical switching ratio of ~ 10 , while this ratio respectively decreases to ~ 8 and ~ 5 when the area decreases to $150\times 150\mu\text{m}^2$ and $100\times 100\mu\text{m}^2$, as shown in Figure R28d. The photoconductance change ratio shows increasing trends with the increased device area since the large area corresponds to a higher volume of the secondary structure change thus a larger photoconductance change. The area-dependent switching is also consistent with the conductance change in the MSFP thin films that arises from the secondary structure change.

The critical intensity is nearly unchanged when changing the device area and the MSFP thin film thickness. The critical intensity for the change from PPM to NPM effect mainly depends on the heat provided by the light¹⁻³. Therefore, for the devices with

different areas or thicknesses, the threshold heat and optical power for turning PPM to NPM are nearly identical. The revision has been added to the revised supplementary information.

Fig. R28. The influences of device area and MSFP thickness on the optical switching behaviours. a-c, The optical switching behaviours in the MSFP memory devices with different MSFP thicknesses of ~63, ~97, and ~186nm, respectively. d, Device area dependent optical LTPs and LTDs in the memory cell with different device areas of 100×100, 150×150 and 200×200 μm^2 .

References

1. Wang, W., et al. An analogue memristor made of silk fibroin polymer. *J. Mater. Chem. C* **9**, 14583-14588 (2021).
2. Min, K., Umar, M., Ryu, S., Lee, S., Kim, S. Silk protein as a new optically transparent adhesion layer for an ultra-smooth sub-10 nm gold layer. *Nanotechnology* **28**, 115201 (2017).
3. Shi, C., et al. New silk road: from mesoscopic reconstruction/functionalization to flexible meso-electronics/photronics based on cocoon silk materials. *Adv. Mater.* **33**, 2005910 (2021).

The corresponding revision in the manuscript is as follows:

“The effects of different MSFP film thicknesses and device areas on the optical PPM and NPM switching are also investigated as shown in Supplementary Figure 15.”

Comment 8:

Image processing in Figure 3: how are images pre-configured and “mapped” onto an array of 12×12 devices? Duration, pulse#? What properties do these images have (e.g., image type/requirements? color/binary? of varying intensities at 1 wavelength)?

Response: Thanks very much for the reviewer’s comments. The 12×12 input image is mapped on the 12×12 MSFP memory array according to 12 different light intensities (12 gray scales) at a constant wavelength of 405nm and a constant pulse number of 50, as shown in Fig. R29. The detail is as follows:

As shown in Figure R29, the image mapping onto a 12×12 device array is conducted by applying the light pulses (pulse number of 50, pulse width of 100 ms, and wavelength of 405nm) to each device in the device array according to pixel intensities in the input image. More specifically, in the mapping process, each pixel corresponds to each device. The input images are grayscale images. Each image has 12 grey scales, corresponding to 12 different light intensities of 40, 45, 50, 55, 60, 70, 75, 80, 85, 90, 95, 100mW. Each input image is composed of two parts: body pattern pixels and background pixels, in which the body pattern pixels with relatively higher brightness corresponds to the higher light intensities of 70, 75, 80, 85, 90, 95, and 100 mW, while the background pixels with lower brightness corresponds to the lower light intensities of 40, 45, 50, 55, and 60mW.

Currently due to the limitations of the optical setup, we input the light stimuli to the devices in the array one by one. However, because of the nonvolatility in our device, the time required for completing the mapping operations would not cause the conductance degradation, which can be proved in our retention test for optical switching, as shown in Fig. R17. The revision has been added to the revised supplementary information.

Fig. R29. The image mapping on the ORRAM model array according to different light intensities (12 grey scales) at a constant wavelength of 405nm and a constant pulse number of 50.

The corresponding revision in the manuscript is as follows:

“A “fish” pattern with background noise is mapped on the 12×12 MSFP-based memory crossbar array, outputting the responsive currents. The image mapping process is illustrated in Supplementary Figure 17.”

Comment 9:

After completing the image preprocessing, the MSFP-based memristor crossbar array could be erased by the light through the NPM effect with zero electrical power consumption.” But was power supplied to the light that you aren’t counting?

Response: Thanks very much for the reviewer’s reminder. The term “zero electrical power consumption” is indeed not suitable since we neglected the laser power consumption. We delete the claim of the “zero electrical power consumption during the image erasing”.

Comment 10:

Are the two banks of MSFP memristors really identical? Identical in composition, #, and dimensions?

Response: Thanks very much for the comment. The two banks of the MSFP memristor array are totally identical with the same fabrication conditions, same composition and dimensions (Fig. R30). The revision has been added to the revised manuscript.

Fig. R30. Optic images of the ORRAM and ERRAM array prepared under the same experiment condition.

Comment 11:

The configuration/operation of the neural network layers is hard to follow: Where is the fully connected network? “Considering the influence of interconnect resistance, the first three row of the memristor array in the rear 12×12 ERRAM mode array are not operated.” What does this mean? Consider adding/improving/labeling additional image(s) to show how information flows through the system. Also, the 3×3 convolution kernel and its use is unclear. Also, what determines the size of this kernel? # of images? number of pixels/devices? images aren't 3×3 . “The convolution kernel adopted here for feature extraction is $[[1/9, 1/9, 1/9], [1/9, 1/9, 1/9], [1/9, 1/9, 1/9]]$ ” How were these values selected/determined?

Response: Thanks very much for the reviewer’s questions.

(1) *The configuration/operation of the neural network layers is hard to follow: Where is the fully connected network?*

Response: The operation of the fully connected layer can be mapped to the 9×6 ERRAM crossbar array as shown in Fig. R31. The last layer in the network is a fully connected layer with 9 input neurons, 3 output neurons and 27 synapses. Since the two differential ERRAM conductances represent a weight value, the fully connected layer can be mapped to a $9 \times 3 \times 2$ (9×6) array section in our memory crossbar array.

(2) *“Considering the influence of interconnect resistance, the first three row of the memristor array in the rear 12×12 ERRAM mode array are not operated.” What does this mean?*

Response: Considering the impact of interconnect resistances, we choose not to use the first three rows of the 12×12 array. The reason is that the first three convolution kernels occupy a 9×6 array area, and the subsequent 9×3 fully connected layer also occupies a 9×6 array area. In total, a 9×12 array area will be occupied, but our array is in a size of 12×12 , leaving a 3×12 array area unused. Placing this unused 3×12 array area in the bottom three rows would result in longer column lines when reading currents, leading to additional line resistance effects. Therefore, the first three rows of the memristor array in the rear 12×12 ERRAM mode array are not operated.

(3) Consider adding/improving/labeling additional image(s) to show how information flows through the system.

Response: We have illustrated the flow of signals and the implementation of the network based on the ORRAM and ERRAM array, as shown in Fig. R31.

A complete CNN neural network consists of one layer of ORRAM convolution, max pooling, ReLU activation function, one layer of ERRAM convolution, max pooling, ReLU activation function, and one fully connected layer, as shown in Figure R31. For the full hardware implementation, the ORRAM array first perceives image information, and then for the in-sensor convolution operation for image feature extraction. The values of the convolution kernels, ranging from -1 to 1, are mapped to a voltage range of -0.2V to 0.2V generated by the DAC and then input into the ORRAM array. The TIA on the column acts as a virtual ground, converting the incoming current into voltage, which is further collected and converted into a digital signal by the ADC. A 2×2 max-pooling operation and ReLU activation function are performed within the ARM core based on these extracted features. Subsequently, these features are further input into ERRAM for convolution and fully connected layer computations. Due to the ReLU activation function, all feature values become positive. When the features are input into the ERRAM array, all feature values are normalized to the range of 0 to 1, and then mapped to the voltage range of 0 to 0.2V. These normalized feature values are

converted into voltage pulses using the DAC and are input into the ERRAM. The convolution operation is performed first, with the TIAs on the columns acting as virtual grounds, collecting the currents on the columns and converting them back into voltages. Afterwards, a 2×2 max-pooling operation and ReLU activation function are executed within the ARM core. Finally, the processed feature values are input into the ERRAM "fully connected layer" to perform classification and output the results. The manuscript has been revised accordingly.

The information flow starts with a 12×12 image, which undergoes in-sensor convolution using a 12×12 ORRAM array for feature extraction, resulting in a $12 \times 12 \times 3$ feature map. This is followed by a 2×2 max pooling layer and ReLU activation function, further reducing the feature map to $6 \times 6 \times 3$. Subsequently, convolution calculations are performed using three 3×3 convolutional kernels with an ERRAM size of 9×6 , resulting in a feature map size of 6×6 , followed by another 2×2 max pooling layer and ReLU activation, yielding a 3×3 feature map. Finally, the feature map is flattened into a one-dimensional vector and sent into a 9×3 fully connected network for classification output implemented with the 9×6 ERRAM array. The supplementary information has been revised accordingly.

Fig. R31. Fully hardware implementation of the neuromorphic visual system based on the multimodal resistive memory arrays with ORRAM mode array for in-sensor pre-processing and the ERRAM mode array for near-sensor high-level processing. **a**, Architecture of the fully hardware-implemented neuromorphic visual system composed of an ORRAM mode array, an ERRAM mode array, a memory array peripheral control system and an STM32 system. **b**, Optical image of the MSFP memory array-based system. **c**, The whole CNN neural network structure implemented with the ORRAM mode array and ERRAM mode array.

The corresponding revision in the manuscript is as follows:

Hardware system workflow

“The image preprocessing algorithms and a convolutional neural network (CNN) are deployed on the hardware system. The 12×12 ORRAM array executes the image preprocessing of contrast enhancement and background denoising and performs the first-layer optical convolutional operation in CNN for image feature extraction. The ERRAM mode array is employed to complete one convolution layer and fully connected layer computations in the CNN. Other network operations, such as max pooling layers and ReLU activation function layers, are executed within the ARM core. (Supplementary Figure 19 and Supplementary Note 7). For the full hardware implementation, the ORRAM array first perceives image information, and then for the in-sensor convolution operation for image feature extraction. The values of the convolution kernels, ranging from -1 to 1, are mapped to a voltage range of -0.2V to 0.2V generated by the DAC and then input into the ORRAM array. The TIA on the column acts as a virtual ground, converting the incoming current into voltage, which is further collected and converted into a digital signal by the ADC. A 2×2 max-pooling operation and ReLU activation function are performed within the ARM core based on

these extracted features. Subsequently, these features are further input into ERRAM for convolution and fully connected layer computations. Due to the ReLU activation function, all feature values become positive. When the features are input into the ERRAM array, all feature values are normalized to the range of 0 to 1, and then mapped to the voltage range of 0 to 0.2V. These normalized feature values are converted into voltage pulses using the DAC and are input into the ERRAM. The convolution operation is performed first, with the TIAs on the columns acting as virtual grounds, collecting the currents on the columns and converting them back into voltages. Afterwards, a 2×2 max-pooling operation and ReLU activation function are executed within the ARM core. Finally, the processed feature values are input into the ERRAM "fully connected layer" to perform classification and output the results."

(4) Also, the 3×3 convolution kernel and its use is unclear. Also, what determines the size of this kernel? # of images? Number of pixels/devices? Images aren't 3×3 .

Response: The 3×3 convolution kernels are mainly used for feature extraction in image processing [1], and these kernels are updated and obtained through back-propagation and training of the neural network (Fig. R32).

The size of the convolution kernel is generally much smaller than that of the image to perform convolution computations efficiently. The purpose of convolution computations is to extract new features from the image, forming a new feature map. Different kernel sizes result in different fields of view in the extracted feature map. For instance, a 5×5 kernel has a wider field of view compared to a 3×3 kernel, but it comes with a higher computational cost. Moreover, two 3×3 kernels can be equivalent to a single 5×5 kernel¹. Therefore, we have used 3×3 convolution kernels. Here the input image is in a size of 12×12 and the kernel size is 3×3 . Initially, a 3×3 pixel patch from

the image is selected, and a dot product sum operation is performed between the 3×3 convolution kernel and the 3×3 pixel patch to obtain a new pixel value. After this calculation, the 3×3 convolution kernel is shifted by one pixel to the right on the image and performs convolution calculations for the entire row, which is followed by being shifted by one pixel downward for the convolution calculations. This process is repeated iteratively to perform convolution calculations across the entire image. The convolution operation based on the ERRAM mode crossbar array is shown in Fig. R32. The feature values extracted by ORRAM are convolved within the ERRAM mode array. The image features, after being processed by ORRAM, are divided into multiple 3×3 patches. These patches are flattened into column vectors and input into ERRAM for the convolution operations. In this configuration, ERRAM functions as the convolutional kernel, with the convolutional kernel being mapped to the conductance values. The input column vectors of the patches are transformed into voltage pulses and fed into ERRAM for efficient multiply-accumulate computations, facilitating convolution calculations.

The formula for convolution computations In ERRAM mode array is as follows:

$$I_N = \sum_{m=1}^9 X_m W_m^N = \sum_{m=1}^9 V_m (G_m^{N+} - G_m^{N-}) = I_N^+ - I_N^-$$

Where X_m represents the value of input data, and W_m^N denotes the weight of the neural network. The weight can be both positive and negative, which is represented using a pair of differential conductances G_m^{N+} and G_m^{N-} . X_m can be mapped to V_m . The V_m and G_m denote the bias voltage applied to the MSFP memory cell and the corresponding conduction value, respectively. I_N is the differential result current between two columns, I_N^+ and I_N^- . The result of the convolution calculation is denoted as I_N . The manuscript has been revised accordingly.

Fig. R32. Convolution implementation in the ERRAM mode array.

Reference

1. Kim, J., et al. Accurate image super-resolution using very deep convolutional networks, *Proceedings of the IEEE conference on computer vision and pattern recognition*. 1646-1654 (2016).

(5) "The convolution kernel adopted here for feature extraction is $[[1/9, 1/9, 1/9], [1/9, 1/9, 1/9], [1/9, 1/9, 1/9]]$ " How were these values selected/determined ?

Response: The convolution kernel used for feature extraction is $[[1/9, 1/9, 1/9], [1/9, 1/9, 1/9], [1/9, 1/9, 1/9]]$. This kernel is manually set as a mean filter, which is a common image processing algorithm used to effectively smooth images, reduce sharpness, and reduce noise ¹. Alternatively, we can train a recognition network online on a computer and then transfer the trained convolution kernel values to our memory crossbar array using weight transfer methods.

Reference

1. P. Coupe, P., et al. An Optimized blockwise nonlocal means denoising filter for 3-D magnetic resonance images. *IEEE Trans. Medic. Imag.* **27**, 425-441 (2008).

Comment 12:

Simulations used in Figure S9 and S10 are not described anywhere.

Response: Thanks very much for this reminder. The corresponding description has

been added to the revised manuscript. Fig. R33 (Figure S9) illustrates the simulated image preprocessing results by the ORRAM mode array for images with larger size. We used Nosiy Albert Einstein photos (457×381) for image preprocessing simulation according to the ORRAM mode device performance characteristics (Fig. R33). Based on the device-level NPM and PPM characteristics, denoising and contrast enhancement of images were achieved. Additionally, based on the in-sensor convolution capability in the ORRAM mode array, the edge feature in the Albery Einstein images can be effectively extracted.

Figures R34a-c (Figures S10a-c) demonstrates the conductance programming capabilities in the 12×12 MSFP ERRAM mode array and the image processing using the MSFP ERRAM array-based hardware. We first randomly programmed a 12×12 ERRAM mode crossbar array, resulting in a random conductance distribution (state 1). State2 shows to the randomly programmed conductance states in the ERRAM mode array using the pulses with input voltage ranging from 0.7V to 0.7V. State3 shows the conductance states in the ERRAM mode array with columns 5, 9, and 10 selected for programming. This demonstrates ERRAM's programmability for high-level image processing. Figures R34d shows the processing of Lenna images through the convolution operations implemented by the ERRAM mode array-based hardware. The convolution kernels set by random pulses are used for image processing, which can result in the various image processing functions such as image blurring and colour inversion of grayscale images, as shown in Figures R34d. The revision has been added to the revised supplementary information.

Fig. R33. Simulations of the preprocessing capabilities of the ORRAM mode array for larger size

images.

Fig. R34. a-c, Conductance programming in the MSFP-based ERRAM mode array. d, Lenna image processing through the convolution operations implemented by the ERRAM mode array-based hardware.

The corresponding revision is following:

“This result also suggests the good programming capability of the MSFP memory array. To demonstrate this point, a series of voltage pulses were randomly input into the ERRAM mode memristor crossbar array to evaluate the weight tuning capability (Supplementary Figure 21a). The ERRAMs can be cyclically and repeatedly programmed into different conductive states, as shown in Supplementary Figures 21b-c. To further demonstrate the convolutional capabilities of the ERRAM array, higher-resolution images are tested. Supplementary Figure 21d presents the feature extraction result of the “Lenna” image after the convolution operation.”

Comment 13:

Power consumption or overall device footprints are not discussed despite suggesting this approach would lead to more compact, efficient systems for image processing.

Response: Thanks very much for the reviewer’s significant suggestions. To examine the potential of this approach, the power consumption and overall device footprints have been discussed.

The footprint of the current single memory cell with current fabrication technology is $200\mu\text{m}\times 200\mu\text{m}$ (Fig. R14) due to the fabrication limitation in the lab, however, the MSFP device can be potentially and easily scaled down to $0.5\mu\text{m}\times 0.5\mu\text{m}$ through aqueous multiphoton lithography (*Nat. Commun.* 2015, 6, 6812). Therefore, to evaluate the potentiality of the system, we adopted the $0.5\mu\text{m}\times 0.5\mu\text{m}$ device area for the potential system performance evaluation.

Limited by the board-level testing system of the printed circuit board (PCB), the readout speed can be less than 25 ns in the peripheral control system of the application-specific integrated circuits (ASICs) at low process nodes (28nm technology node). Therefore, assuming an individual device size of $0.5\times 0.5\mu\text{m}^2$ and a readout speed of 25ns, the chip-level system performance is calculated as shown in Table R2. We also compared our systems with the previously reported system, as shown in Table R4, implying that our system shows a comparable energy efficiency (151.579 TOPS/W) but higher performance density (18.626 TOPS/mm²) compared with the state-of-the-art visual computing systems based on emerging devices. The revision has been added to the revised supplementary information.

Table R2: The system performance based on the ORRAM mode and ERRAM mode array when our memory device is scaled down to $0.5\mu\text{m}\times 0.5\mu\text{m}$.

Performance	$12\times 12\times 2\times 2/25\text{ns}=23.04\text{GOPS}$
Power	$3.8\text{pJ}/25\text{ns}=0.152\text{mW}$
Area	0.001237mm^2
Energy efficiency	$23.04\text{GOPS}/0.152\text{mW}=151.579\text{TOPS/W}$
Performance density	$23.04\text{GOPS}/0.001237\text{mm}^2=18.626\text{TOPS/mm}^2$

Table R4: System performance comparison.

	Ref. 1	Ref. 2	Ref. 3	This work
--	--------	--------	--------	-----------

Memory	Transistor +RRAM	Transistor +RRAM	Transistor +RRAM	ORRAM+ERRAM
Energy efficiency (TOPS/W)	-	75.17	11.014	151.579
Performance density (TOPS/mm ²)	8.5	7.008	1.164	18.626
Area(mm ²)	0.217	0.0263	0.0704	0.001237
Additional CMOS image sensor required	Yes	Yes	Yes	No
In-sensor computing	NO	NO	NO	Yes

The corresponding manuscript has been revised as follows:

“By comparison, this multimodal array-based visual system shows promising potential for future in-sensing neuromorphic computing systems, exhibiting system advantages in terms of integration density and power efficiency compared with the state-of-the-art system (Supplementary Tables 5-6).”

Reference

1. Correll J. M., Jie L., Song S., et al. An 8-bit 20.7 TOPS/W multi-level cell ReRAM-based compute engine, *VLSI Technology and Circuits*, 2022, 264-265.
2. Spetalnick S. D., Chang M., Konno S., et al. A 2.38M cells/mm² 9.81-350 TOPS/W RRAM compute-in-memory macro in 40nm CMOS with hybrid offset/I OFF cancellation and I cell R BLSL drop mitigation, *VLSI Technology and Circuits*, 2023, 1-2.
3. Yao P., Wu H., Gao B., et al. Fully hardware-implemented memristor convolutional neural network. *Nature* 2020, 577, 641-646.

Minor Revision

(1) Page 4. “shortening” not “shorting”

Response: The “shorting” is corrected as “shortening”.

(2) The caption for Figure 1 does not comprehensively explain all parts of the figure. And these are not adequately discussed in the text. Either amend the caption or remove undiscussed content.

Response: According to the suggestion on the Figure 1, we revised the caption of Figure 1 as below:

Fig. 1 | Neuromorphic vision chip based on multimodal resistive memory arrays

with ORRAM mode array for in-sensor image preprocessing and ERRAM mode array for high-level image recognition. An advanced neuromorphic vision system based on MSFP-based multimodal resistive memory arrays with both ORRAM mode for image pre-processing and ERRAM mode for high-level image recognition, simulating the functions of retina cells and visual cortex, respectively. In the biological visual system, the human retina is organized into three nuclear layers and two synaptic layers, in which bipolar cells that include the off and on cells as an inner nuclear layer (INL) connect the outer nuclear layer (ONL, photoreceptors) with the ganglion cell layer (GCL). The images are firstly sensed by the ONL and then are pre-processed by the INL and GCL, and the pre-processed information will be finally transmitted to the visual cortex layer to complete high-level processing. The neuromorphic vision chip consists of an ORRAM mode array with NPM and PPM features for the in-sensor image pre-processing (*e. g.* contrast enhancement and background denoising) and in-sensor convolution for feature extraction, and ERRAM mode array with analogue resistive switching for in-memory high-level image recognition through convolutional neural network (CNN) operations.

(3) *Figure 2b fails to label Pg3 and DHI compounds. And it isn't clear what Sericin is. The use of LRS and HRS terms are confusing for describing the changes in device conductance upon exposure to 405nm light at different intensities. maybe use different labels to differentiate from the ERRAM switching LRS/HRS states.*

Response: The sericin, Pg3 and DHI compounds have been labelled in Figure 2b. The sericin and fibroin are the major components of the natural silk. We revised the LRS and HRS terms in optical switching throughout the manuscript into the PPM states and NPM. The manuscript has been revised accordingly.

(4) *The FTIR spectrum for the MSFP film before light illumination shown in Figure S5 should be plotted in same wavenumber range and y-scale as Figures 2g-h for direct comparison.*

Response: Thanks very much for the suggestion. The Figures 2g-h have been replotted as shown in Figure S5. The manuscript has been revised accordingly.

(5) *Figure 2i isn't mentioned in the text.*

Response: The corresponding Figure 2i description has added in the revised manuscript.

(6) *Poor/confusing wording throughout:*

“...secondary structure outweighs of the decreased β -t (5.77%) secondary structure leading to the PPM effect.” don't both of these changes drive conductivity increase? therefore, one isn't outweighing the other to cause PPM. “The increasing amount β -t secondary structures then becomes higher than that of β -t secondary structures, which leads to an NPM effect.”

Response: To avoid the confused expression, we revised the description accordingly. The β -s secondary structure with layer-by-layer structure contributes to higher conductivity while the β -t secondary structure with turn structure contributes to lower conductivity¹⁻³. Firstly, we define the secondary mole ratio of a specific secondary structure under darkness as “A”, and under specific light illumination defined as “B”. The increasing ratio of a specific secondary structure is $\frac{B-A}{A} \times 100\%$. After the MSFP film was exposed to 80 mW UV light, the mole ratio of β -s obviously increased by 11.70% while the mole ratio of *r-c*, *a-h*, and β -t decreased by 9.86%, 24.0%, and 5.77%, respectively (Figure 2h). Therefore, the increase of the β -s structure (contribute to a higher conductivity), along with the decrease of the β -t (contribute to a higher conductivity) is possibly responsible for the PPM effect observed in the Au/MSFP/Au memristor array. When the light intensity changes from 80 to 40 mW, the mole ratio of the β -s decreases from 59.30% to 55.76% while the mole ratio of β -t increases from 12.41% to 14.51%, causing the NPM effect in the Au/MSFP/Au memory array.”

The corresponding revision has been added to the revised manuscript as follows:

“Interestingly, these secondary structures in the MSFP film can be modulated by light

intensities. After the MSFP film was exposed to 80 mW UV light, the mole ratio of β -s obviously increased by 11.70% while the r -c, a -h, and β -t decreased by 9.86%, 24.0%, and 5.77%, respectively (Figure 2g). When the light intensity changes from 80 to 40 mW, the mole ratio of the β -s decreases from 59.30% to 55.76% while the mole ratio of β -t increases from 12.41% to 14.51% (Figure 2h), causing the NPM effect. The optically induced PPM effect can be mainly due to the increased concentration of beta-sheet (β -s) with layer-by-layer structures that contribute to higher conductivity, while the optically induced NPM effect can be attributed to the increased beta-turn (β -t) or other non-layered structures that contribute to a lower conductivity²⁹⁻³¹, as shown in Figure 2i. Therefore, the increase of the β -s structure (contributing to higher conductivity), along with the decrease of the β -t (contributing to higher conductivity), is possibly responsible for the PPM effect observed in the Au/MSFP/Au memory. It can also be noted that when the UV light intensity changes to 60mW, the mole ratios of the β -s and β -t secondary structures increase by 2.7% and 29.15%, corresponding to an enhanced NPM effect (Supplementary Figure 14b). The comparisons of the changes in these secondary structures under the light illumination with different light intensities (40, 60, and 80 mW) are shown in Supplementary Table 3.”

References

29. Zhang, L., Chen, T., Ban, H. & Liu, L. Hydrogen bonding-assisted thermal conduction in β -sheet crystals of spider silk protein, *Nanoscale* **6**, 7786-7791 (2014).
30. Yun, Y. S., et al, Microporous carbon nanoplates from regenerated silk proteins for supercapacitors, *Adv. Mater.* **25**, 1993-1998 (2013).
31. Min, K., Umar, M., Ryu, S., Lee, S., & Kim, S. Silk protein as a new optically transparent adhesion layer for an ultra-smooth sub-10 nm gold layer. *Nanotechnology* **28**, 115201 (2017).

(7) “The PPM effect presents an intensified growth trend with the pulse number under a higher light intensity.” *Intensified growth = nonlinear increase in conductance?*

Response: Thanks for the reminder. The PPM effect presents a growth trend

instead of intensified growth. The “*intensified*” has been removed from the description.

(8) Pg 15, “*after 50 pulse stimulation*”. *after 200 pulses?* Define all acronyms: ADC, DAC, TIA, ARM, Conclusion: “*high image recognition accuracy of 95%.*” Do you mean 99.5%?

Response: The “*after 50 pulse stimulation*” and “*high image recognition accuracy of 97.3%.*” have been corrected and all acronyms have been defined in revised manuscript.

REVIEWERS' COMMENTS

Reviewer #1 (Remarks to the Author):

I think that authors answered the comments satisfactorily. Hence I recommend it can be published on Nature Communications.

Reviewer #2 (Remarks to the Author):

The authors have sufficiently addressed my questions and prior concerns, which has significantly improved the manuscript. I support its publication in Nature Communications.

Point-by-point Response

We are appreciative of the reviewers' and editors' supportive remarks. We appreciate their time and helpful feedback, which has allowed us to significantly improve our study.

Reviewer #1

I think that authors answered the comments satisfactorily. Hence, I recommend it can be published on Nature Communications.

Response: Thanks very much for the recommendation.

Reviewer #2

The authors have sufficiently addressed my questions and prior concerns, which has significantly improved the manuscript. I support its publication in Nature Communications.

Response: Thanks very much for the recommendation.